# Ancient *Plasmodium* genomes shed light on the history of human malaria

Malaria-causing protozoa of the genus *Plasmodium* have exerted one of the strongest selective pressures on the human genome, and resistance alleles provide biomolecular footprints that outline the historical reach of these species[1]. Nevertheless, debate persists over when and how malaria parasites emerged as human pathogens and spread around the globe[1,2]. To address these questions, we generated high-coverage ancient mitochondrial and nuclear genome-wide data from *P. falciparum*, *P. vivax* and *P. malariae* from 16 countries spanning around 5,500 years of human history. We identified *P. vivax* and *P. falciparum* across geographically disparate regions of Eurasia from as early as the fourth and first millennia BCE, respectively; for *P. vivax*, this evidence pre-dates textual references by several millennia[3]. Genomic analysis supports distinct disease histories for *P. falciparum* and *P. vivax* in the Americas: similarities between now-eliminated European and peri-contact South American strains indicate that European colonizers were the source of American *P. vivax*, whereas the trans-Atlantic slave trade probably introduced *P. falciparum* into the Americas. Our data underscore the role of cross-cultural contacts in the dissemination of malaria, laying the biomolecular foundation for future palaeo-epidemiological research into the impact of *Plasmodium* parasites on human history. Finally, our unexpected discovery of *P. falciparum* in the high-altitude Himalayas provides a rare case study in which individual mobility can be inferred from infection status, adding to our knowledge of cross-cultural connectivity in the region nearly three millennia ago.

Malaria is a vector-borne disease caused by protozoa in the genus *Plasmodium* and is transmitted by female anopheline mosquitoes[4]. It is a major cause of human morbidity and mortality, with an estimated 240 million cases and more than 600,000 fatalities in 2020 (ref. 5). Beyond its current health impact, malaria has profoundly influenced human evolution, exerting one of the strongest identified selective pressures on the human genome. Congenital haematological conditions, including sickle-cell disease, G6PD deficiency and thalassaemia, have persisted because they confer partial resistance to malaria, indicating a long-term relationship between the pathogen and human populations[6].

Of the five primary human-infecting *Plasmodium* species, *P. falciparum* and *P. vivax* account for the vast majority of malaria disease burden today, whereas *P. malariae*, *P. ovale wallikeri* and *P. ovale curtisi* are less common and cause milder symptoms[4]. Previous research indicates that *P. falciparum* emerged through zoonosis from gorillas in sub-Saharan Africa[7]. Date estimates for the most recent common ancestor of extant *P. falciparum* strains range from less than 10,000 to 450,000 years ago[8–10].

The emergence of *P. vivax* is generally considered to pre-date that of *P. falciparum*, but its evolutionary origins are less well understood. Early mitochondrial analyses supported an origin in Southeast Asia, placing *P. vivax* in a clade of *Plasmodium* species infecting macaques and other Southeast-Asian primates[11,12]. Analyses based on nuclear data, including phylogenies and patterns of nucleotide diversity, have provided further support for an Asian origin[13]. However, parasites of the African great apes, notably *P. carteri* and *P. vivax*-like, are now thought to constitute the closest relatives of *P. vivax*[10,14,15]. Together with the near-fixation of the Duffy-negative allele in many human groups in sub-Saharan Africa, this provides strong support for an African origin for *P. vivax*[1]. The Duffy antigen, encoded by the *FY* locus, facilitates *P. vivax* erythrocyte invasion, and individuals homozygous for the Duffy-negative allele were once considered completely immune to *P. vivax* malaria[1,6]. Accumulating evidence demonstrates that populations with high rates of Duffy negativity can maintain low levels of *P. vivax* transmission, and the phenotype seems to reduce the efficiency of erythrocyte invasion and provide protection against blood-stage infection[16]. Thus, proponents of the African-origin hypothesis argue that a long history of selection pressure exerted by *P. vivax* drove increases in the Duffy-negative phenotype, making these populations less susceptible to *P. vivax* infection today. Interestingly, some human groups in Papua New Guinea have a Duffy null allele that seems to have arisen through an independent mutation. Indeed, the low frequency and long haplotype associated with the Papua New Guinea variant support more recent positive selection in people living in Oceania than in those in sub-Saharan Africa[17].

As well as the evolutionary constraints, variation in pathogenesis between *P. vivax* and *P. falciparum* contributes to their distinct geographical distributions and ecologies. Because of its higher virulence, morbidity and mortality, *P. falciparum* requires a larger population of susceptible hosts to sustain transmission. Consequently, some researchers have theorized that hunter-gatherer population densities were probably too low to support the emergence of *P. falciparum*, which instead may have proliferated with the development of agriculture in sub-Saharan Africa[1]. Climate also poses distinct constraints on the

ranges of these two species, with *P. vivax* able to survive and develop at lower temperatures than *P. falciparum*[18,19]. Finally, *P. vivax* forms hypnozoites in its dormant hepatic stage, and reactivation months or even years after an initial infection can re-initiate the *Plasmodium* life cycle, enabling further transmission[4]. Hypnozoites enable *P. vivax* to overwinter in the human host when low temperatures limit vector activity. Combined with its greater tolerance for cold temperatures, this capacity enables *P. vivax* to survive in temperate regions, whereas *P. falciparum* is generally restricted to tropical and subtropical zones[1].

Because *Plasmodium* species are obligate intracellular pathogens, their contemporary distributions reflect patterns of human mobility, as well as the evolutionary, physiological and ecological constraints acting on the parasite, human host and mosquito vector. However, relatively little is known about the timing and routes by which *Plasmodium* spp. spread around the globe. In the palaeopathological literature, cribra orbitalia and porotic hyperostosis have been considered to be indicators of severe malarial anaemia[20,21]. However, their presence should be interpreted with caution because these skeletal lesions are not pathognomonic for the identification of malaria cases in the archaeological record[22,23], and the two conditions probably have different underlying aetiologies[24,25]. Recurrent fevers are described in Vedic and Brahmanic texts from the first millennium BCE, and Hippocratic texts from the late fifth or early fourth century BCE provide the first unambiguous references to malaria in the Mediterranean world[1,3]. However, retrospective diagnosis of malaria poses considerable challenges, and many time periods and regions are missing from the historical record[26]. Although written sources and congenital haematological conditions provide indirect evidence of the historical range of malaria, uncertainty persists over which species contributed to selective processes in specific regions, as well as how the selective dynamics played out over time[1,2].

Tracing the history of *Plasmodium* spp. in the Americas is of particular interest, given the limited number of transoceanic contacts that may have facilitated transmission. *P. falciparum* is likely to have reached the Americas with colonizers from Mediterranean Europe or as a result of the trans-Atlantic slave trade, but the potential pre-contact origin of American *P. vivax* is still debated[27]. Some scholars suggest that *P. vivax* reached the American continent with its first human inhabitants, and cite as evidence both its high nucleotide diversity and the presence of divergent mitochondrial lineages in American parasite populations[28]. Others argue that American *P. vivax* may derive from pre-colonial-era contacts with Oceanian seafarers[27]. Finally, *P. vivax*, as well as *P. falciparum* and many other Eurasian pathogens, may have reached the Americas during the European colonial era[28–30]. A contact-era introduction of *Plasmodium* spp. is consistent with the absence of malaria-resistance alleles in the Indigenous peoples of the Americas[31]. Further support for this hypothesis comes from analyses of the only historical European *P. vivax* genomic dataset available to date, which derives from a 1944 blood slide from Spain's Ebro Delta. Analysis of nuclear single-nucleotide polymorphism (SNP) data places Ebro1944 close to contemporary South and Central American *P. vivax* strains[30].

The ability to retrieve ancient bacterial and viral DNA preserved in human skeletal material is providing a fuller picture of the evolution, origins and global dissemination of historically important pathogens[32]. However, attempts to retrieve ancient DNA from *Plasmodium* spp. have until now had limited success[33]. Apart from Ebro1944 (refs. 30,34,35), the available ancient *Plasmodium* datasets have so far been restricted to two partial mitochondrial genomes from southern Italy dating to the first and second century CE[36]. Here we identify *P. falciparum*, *P. vivax* and *P. malariae* infections in 36 ancient individuals from 16 countries spanning 5,500 years of human history from the Neolithic to the modern era. Using two new in-solution hybridization capture bait sets, we generate high-coverage ancient *Plasmodium* mitochondrial genomes and genome-wide nuclear data, which demonstrate that the European expansion of *P. vivax* greatly pre-dates evidence from written sources. Genomic data from now-eliminated European *P. falciparum* and *P. vivax*

strains provide an unprecedented opportunity to explore gaps in the genomic diversity of modern *Plasmodium* populations, enabling a fuller picture of the origins and transmission routes of human malaria parasites. Finally, contextualizing ancient genomic data from *P. falciparum* and *P. vivax* alongside archaeological information and human population genetics reveals the critical role of human mobility in the spread of malaria in past populations.

## Ancient *Plasmodium* spp. data generation

To identify ancient malaria cases, we performed a metagenomic analysis of previously produced shotgun-sequenced libraries from more than 10,000 ancient individuals (Methods). Ancient DNA libraries found to possess traces of *Plasmodium* DNA were enriched using two new hybridization capture reagents targeting the mitochondrial and nuclear genomes of *Plasmodium* spp. In total, we identified 36 malaria cases, comprising 10 *P. falciparum* infections, 2 cases of *P. malariae* and 21 *P. vivax* infections, along with 2 individuals co-infected with *P. falciparum* and *P. malariae* as well as 1 *P. vivax*–*P. falciparum* co-infection (Fig. 1, Supplementary Table 1 and Supplementary Note 1). We analysed these ancient mitochondrial and nuclear datasets alongside modern *Plasmodium* data and published shotgun reads from the Ebro1944 blood slide[30,34,35,37,38].

Mitochondrial capture allowed for the reconstruction of full genomes from 13 *P. falciparum* strains with mean coverage ranging from 1.1× to 118.3×, 6 *P. vivax* strains with mean coverage of 3.0× to 94.3× and 4 *P. malariae* strains with 1.1× to 80.4× mean coverage (Extended Data Fig. 1, Supplementary Table 2 and Supplementary Note 2). To further explore the population structure of ancient *P. vivax* and *P. falciparum*, we genotyped our ancient nuclear-capture datasets at high-quality biallelic SNP positions ascertained in modern datasets published by the MalariaGEN *P. vivax* Genome Variation Project and the MalariaGEN *P. falciparum* Community Project, respectively[37,38] (Extended Data Fig. 2). For *P. falciparum*, we merged data from 1,227 modern and 8 ancient strains genotyped at 106,179 segregating SNP positions, and for *P. vivax* our final dataset contained 906 modern and 23 ancient strains genotyped at 419,387 segregating SNP positions. The coverage of our ancient samples ranged from 541 to 19,525 SNPs for *P. falciparum* (median of 1,068 SNPs) and from 721 to 208,344 SNPs for *P. vivax* (median of 2,153.5 SNPs) (Supplementary Table 3 and Supplementary Note 3).

## Early presence of malaria in Eurasia

Previous attempts to outline the past distribution of *Plasmodium* spp. have relied on textual references that provided evidence for *P. falciparum* in the Greek world as early as around 400 BCE and in South Asia from the early first millennium BCE[3]. Our ancient *P. falciparum* data from the Himalayan site of Chokhopani (a calibrated (cal) date of around 804–765 cal BCE[39]; Supplementary Note 1.1.3) and the Central European Iron Age site of Göttlesbrunn (around 350–250 BCE; Supplementary Note 1.1.6) complement these textual references, shedding light on the role of mobility and trade in transmitting malaria beyond historically documented centres of endemicity (Fig. 1). Chokhopani is situated in a high transverse Himalayan valley around 2,800 m above sea level, but grave goods indicate that there were trade connections with the Indian subcontinent that may have facilitated the spread of malaria into the highlands[39]. Similarly, Göttlesbrunn was part of some trans-regional exchange networks, as evidenced by the archaeological record[40], and historically attested conflicts brought Late Iron Age Central European populations into potentially malarious regions of the Mediterranean and the Balkans[41].

Biomolecular data also provide firm evidence for the widespread impact of *P. vivax* on prehistoric European populations. We have identified three *P. vivax*-infected individuals dating from the third or fourth millennium BCE, including a Middle Neolithic Baalberge individual from

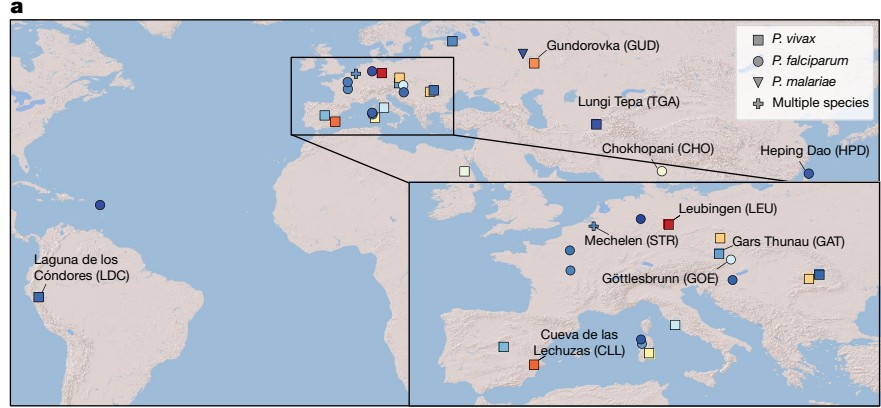

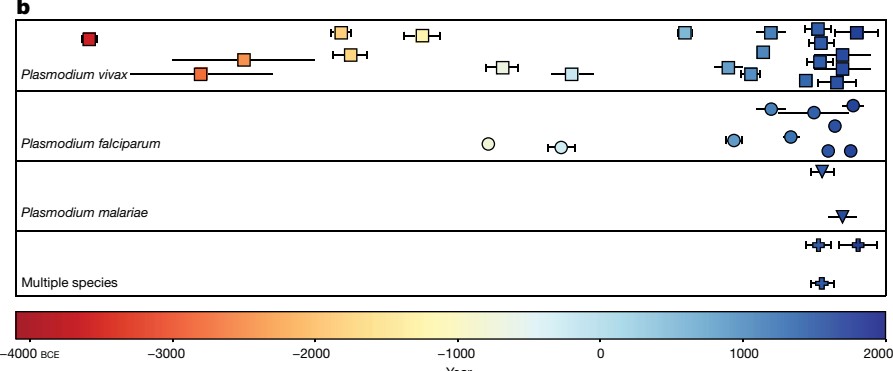

**Fig. 1 | Spatial and temporal distribution of *Plasmodium*-positive ancient individuals. a**, Archaeological sites with malaria-positive ancient individuals. Site colour reflects the date-range midpoint for the infected individual(s). Names and abbreviations are included for sites discussed in the main text. Map produced using Cartopy (v.0.20.3, https://github.com/SciTools/cartopy/tree/ v0.20.3), Natural Earth (naturalearthdata.com) and World Shaded Relief map (Esri). **b**, Temporal distribution of *n* = 36 malaria-positive ancient individuals. Points reflect date-range midpoints; error bars indicate uncertainty inferred from either archaeological context (uncapped error bars) or radiocarbon dating (capped error bars, calibrated calendar ages, 2σ range) (Supplementary Table 1).

Leubingen, Germany (3,637–3,528 cal BCE; Supplementary Note 1.2.7), a Chalcolithic individual from Cueva de las Lechuzas, Spain (3,300–2,300 BCE[42]; Supplementary Note 1.2.1) and an Eneolithic individual from Gundorovka in Russia (turn of the fourth to third millennium BCE[43]; Supplementary Note 1.2.5) (Fig. 1). Finding *P. vivax* in 3 ecologically disparate sites more than 5,000 km apart indicates that this species probably affected large portions of Europe by the fourth millennium BCE, predating the earliest textual evidence for malaria by several thousand years[1,3]. Evidence for *P. vivax* infection at Gundorovka is especially noteworthy: although the site was used for a period spanning the Neolithic–Eneolithic through the Middle–Late Bronze Age and Early Iron Age, the individual analysed here has been contextually dated to the Eneolithic period[43]. Our findings underscore the need for further sampling to fully elucidate the capacity of low-density transitional hunter-gatherer groups to sustain malaria transmission before the full-scale adoption of agriculture and sedentism.

## *P. vivax* population genetics

Consistent with previous studies, analysis of nuclear SNP data revealed a strong phylogeographic structure in modern *P. vivax* populations[37]. In a principal component analysis (PCA), strains from proximal regions formed distinct clusters, and the first two principal components (PCs) captured a large proportion of this genetic variation (9.47% and 5.54% for PC1 and PC2, respectively), defining three main clusters: (1) Africa, Western Asia and Latin America (South and Central America); (2) East and Southeast Asia; and (3) Oceania (Fig. 2). Our ancient *P. vivax* dataset includes six strains with nuclear SNP coverage levels suitable for population genetic analysis (Supplementary Note 4): STR105 and STR185

from the medieval/early modern cemetery of St. Rombout in Mechelen, Belgium (Supplementary Note 1.4.1); GAT004 from the early medieval Austrian site Gars Thunau (Supplementary Note 1.2.3); the previously published Ebro1944 dataset[30,34,35]; LDC020, dated to the peri-contact period (1437–1617 cal CE) from the Chachapoya site of Laguna de los Cóndores, Peru (Supplementary Note 1.2.6); and TGA007 from the late medieval/early modern period in southern Uzbekistan (Supplementary Note 1.2.8). Our data provide an opportunity to assess diversity in European *P. vivax* populations spanning the colonial era. All higher-coverage European strains fall in a tight cluster in PCA space, indicating the presence of a single, broadly distributed European population exhibiting genetic continuity from the medieval to the modern period (Fig. 2). Assessment of our ancient samples using PCA, ADMIXTURE and $F_4$ statistics also provided evidence for stability in *P. vivax* population structure over time (Extended Data Figs. 3–6, Supplementary Table 4 and Supplementary Note 5). Falling within the diversity of modern Latin American strains, LDC020 exhibits a closer affinity to modern Peruvian *P. vivax* than to modern strains from Colombia, Brazil and Central America (Supplementary Table 5 and Supplementary Note 6). Similarly, PCA places TGA007 adjacent to modern Western Asian populations sampled from Afghanistan, India, Iran and Sri Lanka, and adjacent to and shifted towards two admixed strains from modern Bhutan. Finally, low-coverage samples from Uzbekistan and Pharaonic Egypt also show relatedness to geographically proximal modern populations (Extended Data Fig. 3 and Supplementary Note 5). Such affinities in strains sampled centuries apart may reflect long-term persistence of endemic foci in Latin America and western/southern Asia, an observation that is consistent with the refractory nature of *P. vivax* populations to contemporary eradication campaigns[44].

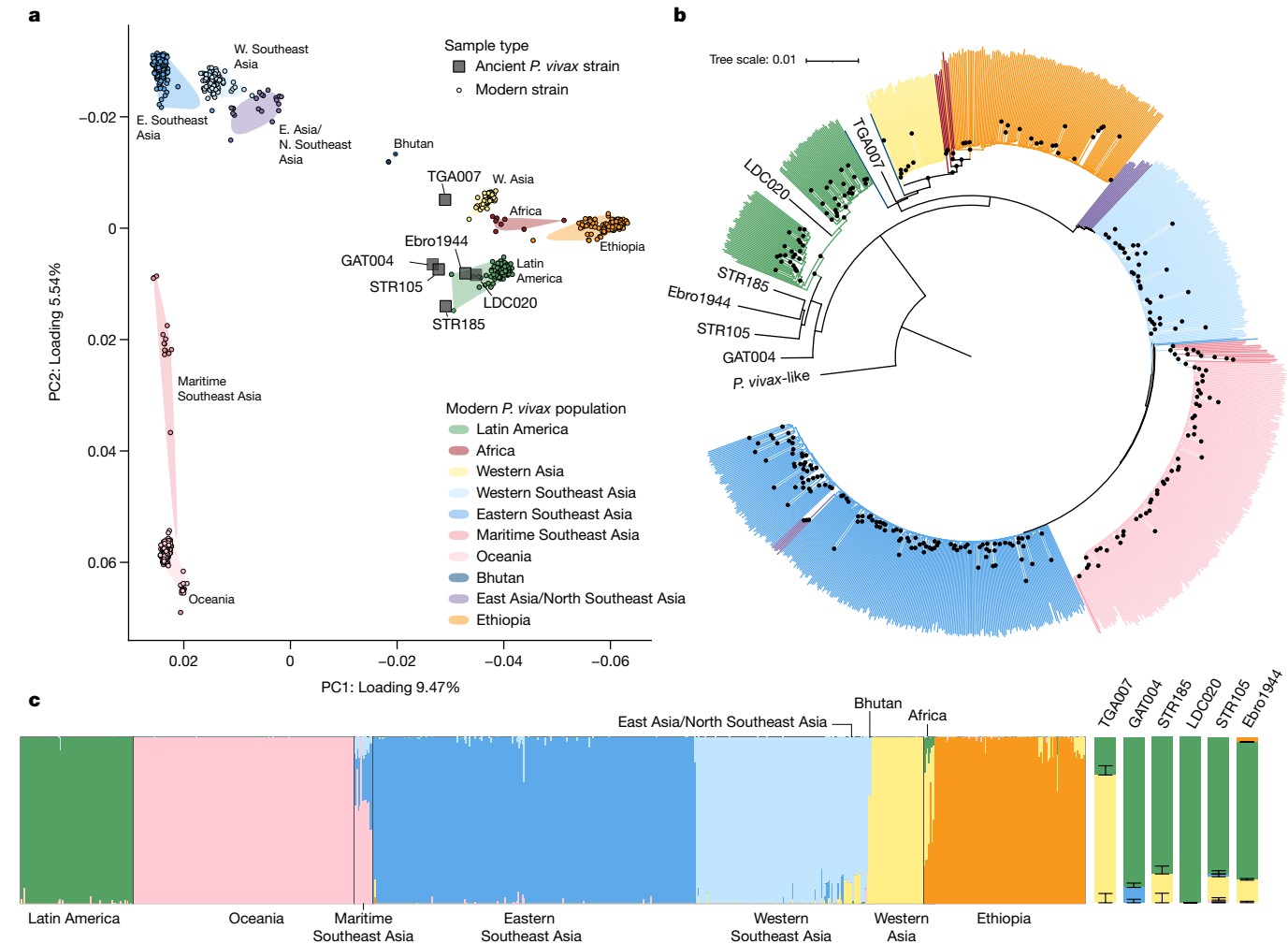

**Fig. 2 | *P. vivax* population genetics. a**, Ancient *P. vivax* strains with more than 5,000 SNPs covered (grey squares). Ancient data are projected onto modern *P. vivax* strains (small points) genotyped by the MalariaGEN *P. vivax* Genome Variation Project[37]. Shaded regions delimit the spread of modern *P. vivax* populations in PCA space. **b**, Neighbour-joining phylogeny including ancient and modern *P. vivax* strains. Branches are coloured by geographic origin, as in **a**, black points reflect nodes receiving support values greater than or equal to 0.9 (100 bootstrap replicates). **c**, Unsupervised ADMIXTURE analysis of modern *P. vivax* populations using $K = 6$ ancestry sources (left), and supervised ADMIXTURE analysis of high-coverage (more than 5,000 SNPs) ancient *P. vivax* strains (right). Ancient strains were modelled as mixtures of $K = 6$ ancestral sources maximized in the following modern populations: Latin America, Oceania, maritime Southeast Asia, eastern Southeast Asia, western Southeast Asia, western Asia and Ethiopia. Error bars reflect uncertainty in mean individual admixture proportions (standard errors, 300 bootstrap replicates).

## *P. falciparum* population genetics

As for *P. vivax*, analysis of nuclear SNP data revealed considerable phylogeographic structure in modern *P. falciparum* populations, with PCA defining the following three clusters: (1) Africa and South America; (2) South and Southeast Asia; and (3) Oceania (Fig. 3). As previously observed, modern *P. falciparum* exhibits lower genetic diversity than *P. vivax*. In a global set of 1,227 *P. falciparum* samples published by the MalariaGEN project[38], we observed only 106,179 high-quality biallelic segregating SNPs, compared with 419,387 positions in a set of 906 *P. vivax* strains. Furthermore, as a consequence of the organism's higher AT skew and lower complexity, our probe set spans a smaller proportion of the *P. falciparum* nuclear genome (Extended Data Fig. 7), meaning that *P. falciparum* strains generally attain lower coverage in our ancient dataset. Nevertheless, 3 samples exhibit coverage levels of more than 10,000 SNPs: CHO001 from the first millennium BCE Himalayan site of Chokhopani (Supplementary Note 1.1.3); HPD007 from the seventeenth-century Spanish colonial outpost of Heping Dao off the coast of Taiwan (Supplementary Note 1.1.7); and Ebro1944 (refs. 30,34,35). Interestingly, these genomes, along with other lower-coverage European strains, fall into a gap in PCA space and are modelled as complex population mixtures in supervised ADMIXTURE analysis (Fig. 3, Extended Data Figs. 4, 5 and 8 and Supplementary Note 7). This observation indicates that our ancient strains cannot be clearly assigned to one currently sampled modern *P. falciparum* population, possibly reflecting sampling biases in modern comparative datasets. Apart from Ebro1944, our ancient dataset provides a first glimpse into the genetics of now-eliminated European *P. falciparum* populations. Furthermore, despite constituting an important centre of *P. falciparum* endemicity, the MalariaGEN *P. falciparum* Community Project Pf6 data release lacks genotype data from India. We attempted to address this problem by analysing our data alongside published shotgun-sequencing data from five *P. falciparum* strains retrieved from hospitalized patients in Goa[45]. Indeed, based on PCA, $F_3$ statistics and ChromoPainter/fineSTRUCTURE, the Ebro1944 strain showed a higher affinity to these Indian genomes than to other modern populations (Fig. 3, Extended Data Fig. 8 and Supplementary Note 7). This observation may reflect links between European and South Asian *P. falciparum* populations, as previously proposed[34], but more sampling is needed to further support this hypothesis and clarify the population affinities of our ancient strains.

## Alternative histories in the Americas

In this study, we generated high-coverage *P. vivax* genome-wide nuclear and mitochondrial data from a peri-contact South American individual from the site of Laguna de los Cóndores in Peru (LDC020). Associated with the Chachapoya culture and radiocarbon dated to between 1437 and 1617 cal CE, analysis of human genome-wide data indicated an individual of Indigenous ancestry with no evidence of European admixture (Extended Data Fig. 9 and Supplementary Note 8). The LDC020 *P. vivax* strain overlaps with modern South American populations in PCA and can be modelled as deriving 100% of its genetic ancestry from American-related populations in supervised ADMIXTURE analysis (200 bootstrap replicates; Fig. 2). $F_3$ statistics indicate that LDC020 is related more closely to Latin America than to any other modern *P. vivax* population (Extended Data Fig. 3 and Supplementary Note 5), and $F_4$ statistics demonstrate that LDC020 shows excess affinity with modern Peruvian *P. vivax* populations compared with modern strains from Colombia, Brazil and Central America (Supplementary Table 5 and Supplementary Note 6). Together, this evidence suggests that LDC020 is closely related to the ancestors of *P. vivax* circulating in the Americas today, and the genetic links between modern and ancient Peruvian *P. vivax* support the early establishment and long-term maintenance of an endemic focus in the region.

Interestingly, both PCA and $F_4$ statistics indicate that ancient European *P. vivax* strains are also related more closely to modern and ancient Latin American strains than to any other modern population (Fig. 2, Extended Data Fig. 3, Supplementary Table 4 and Supplementary Note 5). A neighbour-joining phylogeny constructed using genome-wide SNP data places the ancient European *P. vivax* strains basal to a clade formed by LDC020 and modern Latin American lineages (Fig. 2). Together, the close relationship between pre-elimination European populations, modern American *P. vivax* and LDC020 supports the introduction of *P. vivax* from European populations to the Americas during the contact period. A non-African source for American *P. vivax* is also consistent with the low frequency of *P. vivax* in regions of sub-Saharan Africa with high rates of Duffy negativity. Overall, this evidence for a close genetic link between American and extirpated European strains indicates that *P. vivax* was probably absent in the Americas before the colonial period, although we cannot exclude the possibility of a replacement of pre-contact *P. vivax* variation after the introduction of strains from Europe.

Although our dataset lacks ancient Latin American *P. falciparum* strains, it sheds light on the relatedness between modern lineages and ancient European *P. falciparum* strains spanning the contact era. As noted above, ancient European *P. falciparum* strains fall in a distinct region in PCA space that does not overlap with currently sampled modern populations. On the contrary, all modern South American *P. falciparum* strains sequenced to date form a tight cluster closely related to strains from West, Central and East Africa. Analyses using $F_4$ statistics further support the close relationship between South American and African *P. falciparum*, although a minor contribution from European lineages cannot be excluded (Supplementary Table 6 and Supplementary Note 7). Together with the high prevalence of *P. falciparum* in sub-Saharan Africa today, our population genetic analysis supports the transmission of this species to the Americas as a result of the trans-Atlantic slave trade[27,46].

## Human mobility and malaria transmission

The unexpected recovery of *P. falciparum* and *P. vivax* genomes from individuals at the high-altitude Himalayan site of Chokhopani (2,800 m above sea level) and the Andean site of Laguna de los Cóndores (2,860 m above sea level) underscores the role of human mobility in spreading malaria. In general, elevation limits endogenous malaria transmission. The colder and potentially drier conditions associated with high

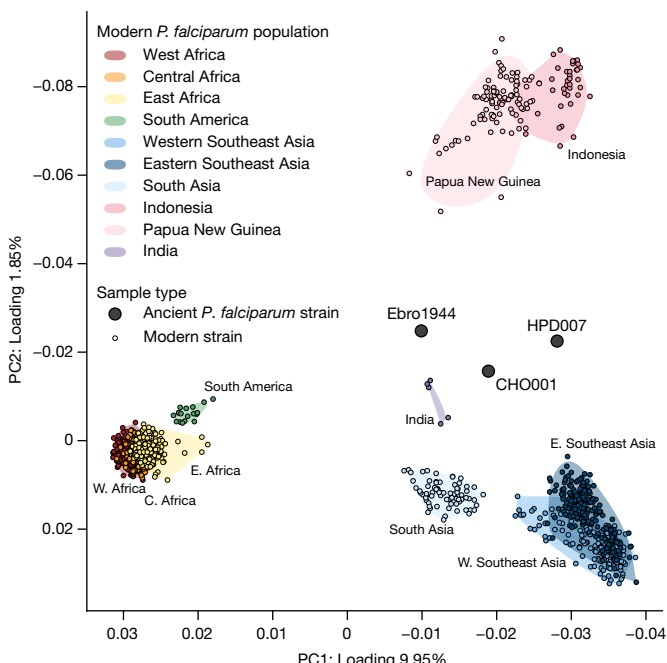

**Fig. 3 | *P. falciparum* PCA.** Ancient *P. falciparum* strains with more than 10,000 SNPs covered (labelled circles). Ancient strains are projected onto the diversity of modern *P. falciparum* genomes published by the MalariaGEN *P. falciparum* Community Project[38]. Modern strains are shown as small points, and the shaded regions delimit the distribution of modern *P. falciparum* populations in PCA space.

altitudes may be unsuitable for mosquito survival and reproduction, and temperatures below species-specific thresholds inhibit the development of *Plasmodium* parasites inside mosquito vectors[47]. Precise altitudinal limits on malaria endemicity depend on a variety of factors, including latitude, microclimate, landscape modification and the *Plasmodium* and *Anopheles* species present, and boundaries may shift dynamically in response to changes in climate and/or the local environment. Although the complex ecology of *Plasmodium* transmission complicates attempts to reconstruct past endemic ranges, modern epidemiological and climatological data are sufficient to render malaria transmission at Chokhopani highly unlikely (Supplementary Note 9).

Instead, we hypothesize that malaria cases at highland sites reflect transregional transmission from lowland areas capable of sustaining endemic foci. Situated in a high transverse Himalayan valley linking the Tibetan Plateau with southern lowland areas, the region surrounding Chokhopani may have served as an epicentre of trade and exchange in the first millennium BCE. Consisting of a series of shaft tombs built into a riverside cliff, the site contained three burial chambers containing the remains of at least 21 individuals, as well as copper grave goods similar to those produced in the Indian subcontinent[39,48,49] (Supplementary Note 1.1.3). Owing to the commingled nature of the remains, skeletal material from CHO001 was limited to the permanent molar yielding *P. falciparum* DNA. Previous studies found that the genetically male individual CHO001 possessed alleles associated with high-altitude adaptation and exhibited ancestry similar to that of present-day Tibetans[50] (Supplementary Note 1.1.3). Notably, individuals from Chokhopani also have a minor lowland South Asian ancestry component that is absent in other prehistoric sites in Upper Mustang; this finding further supports the connection between Chokhopani and lowland South Asian regions, although the admixture event probably occurred around 500–1,000 years before the *P. falciparum*-infected individual identified here lived[50]. Finally, the relatively short overland distances between Chokhopani and regions of contemporary malaria endemicity in the Nepalese and

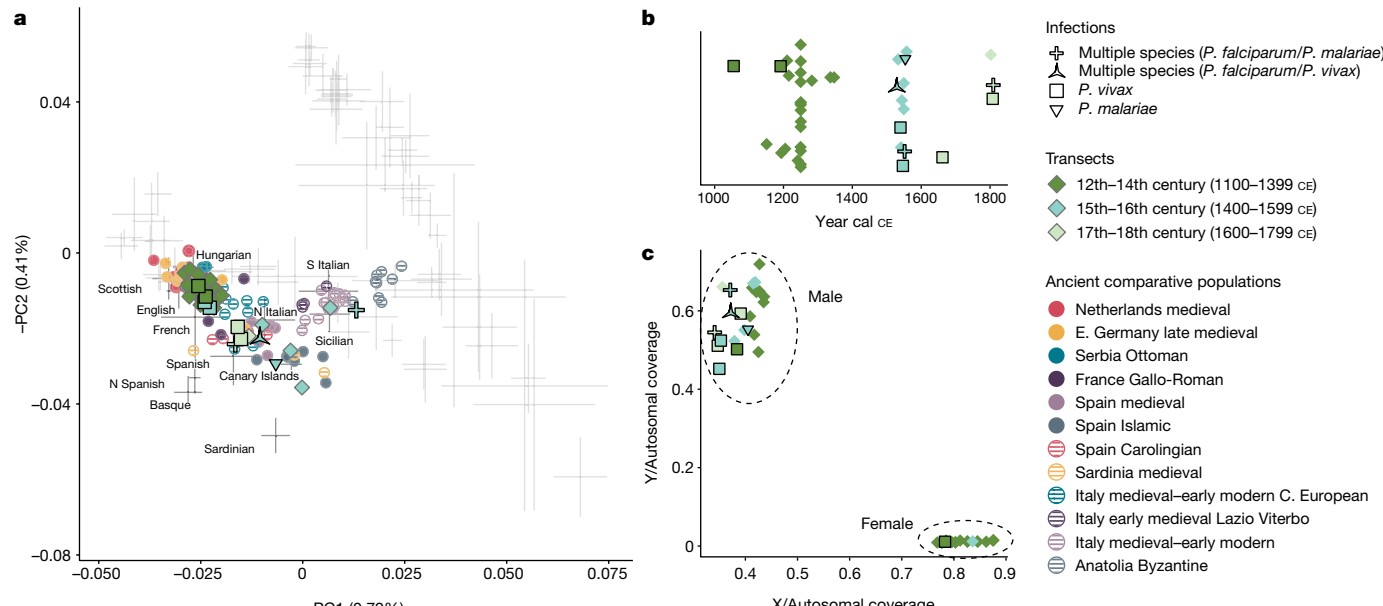

**Fig. 4 | Shift in human ancestry and malaria infectivity at Mechelen, Belgium. a**, PCA showing both infected and uninfected ancient individuals projected onto the diversity of modern Western Eurasian populations. Marker type indicates infection status and colouration reflects the temporal layer of ancient individuals. Selected ancient populations are shown as coloured circles for comparative purposes. **b**, Chronology of individuals yielding human and/or *Plasmodium* genome-wide data. Individuals are classified as deriving from the twelfth-to-fourteenth centuries CE, the fifteenth-to-sixteenth centuries or the seventeenth-to-eighteenth centuries on the basis of their calibrated radiocarbon dates or available archaeological context. **c**, Relative coverage on the X and Y chromosomes used for sex determination.

Indian Terai underscore the likely role of individual mobility in spreading *P. falciparum* into the Himalayan highlands[51]. Taken together, our discovery of a *P. falciparum* infection in the Chokhopani individual adds to a growing body of evidence for cross-cultural connectivity, even in this remote Himalayan region. Given the genetic links between CHO001 and other modern and ancient high-altitude populations, we suggest that this individual lived locally and contracted malaria while travelling to or from an adjacent endemic region. However, we cannot exclude the possibility that CHO001 was a non-local individual who travelled to Chokhopani from a nearby malarious area. Overall, we highlight CHO001 as a rare case study in which aspects of an individual's mobility can be inferred from their infectious-disease status, which is an important finding given the limited information that could be drawn from the fragmented skeletal material associated with this individual.

Long-distance exchange may also have facilitated the spread of *P. vivax* into the vicinity of Laguna de los Cóndores (LDC; Supplementary Note 9). The Chachapoya cultural region, including LDC, is in the subtropical forest of the eastern Andean slopes, providing an appropriate environment for mosquitos to thrive. Despite the remote location of the region today, archaeological evidence suggests that the Chachapoya cultural region was home to many pre-colonial societies and served as an intersection of cultural connectivity and exchange for communities across the Andes to the Amazon Basin[52]. Indeed, the discovery of Amazonian feathered head-dresses and preserved lowland-animal pelts at LDC attests to exchange networks with areas of modern malaria endemicity[52]. Furthermore, the Spanish invasion and conquest is known to be one of the main factors contributing to the spread of infectious diseases throughout the Americas, leading to drastic population declines for many Indigenous groups that some suggest were as large as 90% (ref. 53). In some regions, introduced pathogens spread rapidly along existing networks of connectivity, decimating local Indigenous populations even before the arrival of colonial military forces[53,54]. Later, the Spanish displaced large numbers of Indigenous inhabitants, who were conscripted to fight against the Inca or to explore the Amazon[55]. Together, warfare, Spanish colonization

and other socio-political upheavals may have accelerated the spread of malaria in the Andean hinterlands early in the colonial era.

The identification of ten malaria-infected individuals from the cemetery of St. Rombout in Mechelen, Belgium, further illustrates the capacity of warfare and individual mobilization to drive malaria transmission (Supplementary Note 1.4.1). Situated directly adjacent to the first permanent military hospital in early modern Europe, which was in use from 1567 to 1715 CE, the cemetery may have served as a burial place for soldiers in the Habsburg Army of Flanders[56,57]. Excavations of the cemetery unearthed the remains of 4,158 articulated individuals from 3 main layers, approximately dated to the twelfth-to-fourteenth centuries CE, the fifteenth-to-sixteenth centuries CE and the seventeenth-to-eighteenth centuries CE; the last 2 phases overlap with the time the hospital was in use[57,58]. Interestingly, our pathogenomic and human population genetic analyses of 40 individuals from Mechelen support the hypothesis that the cemetery contained at least 2 distinct subgroups. Studying 25 individuals dated to the earliest phase (twelfth-to-fourteenth centuries CE) reveals an approximately equal sex ratio, and these individuals formed a tight cluster in PCA overlapping with geographically proximal modern populations for which genotype data are available (including French, English, Scottish and Hungarian), as well as late-medieval Germany and the Netherlands[59] (Fig. 4 and Supplementary Table 7). Consistent with this signature of central/northern European ancestry, both of the malaria infections in the early transect were caused by *P. vivax*, a species adapted to transmission in colder climates and thought to be endemic throughout Europe at this time[26].

Compared with the early transect, 15 individuals recovered from the cemetery's middle and late phases exhibit greater variability in both genetic ancestry and *Plasmodium* species detected. Of the 13 male individuals, 11 have heterogeneous ancestry encountered across the Mediterranean, and 2 female individuals overlap the early phase cluster in PCA space (Fig. 4 and Supplementary Table 7). Interestingly, we identified *P. vivax*, *P. malariae* and/or *P. falciparum* in eight mid–late-phase male individuals, including three cases of multispecies *Plasmodium* infections, which are common today in geographic regions with more

than one endemic species[60]. To refine the possible source populations for these eight later-phase infected individuals, we performed further analyses using tools for ancestry spatial interpolation and modelling (Supplementary Note 10). As in the early phase, two *P. vivax*-infected individuals had ancestry similar to populations from central/northern Europe, consistent with a 'local' ancestry signature. For the remaining 'non-local' malaria cases, we narrowed down the possible sources to the southern Iberian peninsula (*n* = 3) and the Aegean (*n* = 1), and in two cases our modelling indicated mixed ancestry including both these former sources and Sardinia (Extended Data Fig. 10, Supplementary Table 8 and Supplementary Note 10). Remarkably, all individuals infected with *P. falciparum* and/or *P. malariae*, including the three individuals with multispecies *Plasmodium* infections, exhibited non-local ancestry. Low winter temperatures are thought to have restricted endemic *P. falciparum* foci north of the Alps[26], but these findings are consistent with the hypothesis that the mid–late-phase malaria-infected individuals from Mechelen may have been troops from the circum-Mediterranean region. More broadly, our results are consistent with the historical records regarding the army of Flanders, which in the sixteenth and seventeenth centuries CE recruited soldiers from northern Italy, Spain and other Mediterranean regions to fight in the Low Countries[61]. As well as providing compelling evidence regarding the mortuary context of these individuals, the host and pathogenic DNA retrieved raises important questions regarding the extent of local malaria outbreaks in this period. Notably, multiple anopheline vectors capable of transmitting *P. falciparum* and other malaria parasites persist in the Low Countries and other regions of Europe today[26,62]. Thus, although *P. falciparum*-infected individuals at Mechelen may represent isolated, recently imported cases, it is also possible that they fell victim to more-extensive local malaria outbreaks triggered by intense human mobilization in the socio-economic context of warfare.

## Conclusions and implications

In this study, we demonstrate that malaria-parasite genome-wide mitochondrial and nuclear data can be reconstructed from human skeletal remains. Together with textual, osteological and archaeological evidence, these new biomolecular data provide an opportunity to reassess our understanding of the past distribution of malaria-parasite species. We show that *P. vivax* was endemic in Europe several thousand years before the earliest textual references, and the identification of *P. falciparum* in the Himalayan highlands and temperate Europe underscores the role of human mobility in carrying malaria to the peripheries of endemic zones. As well as species identification, we demonstrate that population genetic analysis of unsampled and eliminated parasite populations can provide critical insights into the sociocultural processes that helped to spread malaria around the globe. We find that eliminated European *P. vivax* resembles modern and ancient Latin American parasite populations, consistent with transmission from European colonizers to Indigenous peoples of the Americas in the contact period. We also found that American *P. falciparum* shows strong affinity to modern African lineages, implicating the trans-Atlantic slave trade in the spread of this parasite across the Atlantic.

Beyond these insights, the capacity to reconstruct ancient genomes from human malaria parasites raises new questions and opens multiple avenues for future research. The population history of European *P. falciparum* remains particularly enigmatic, with ancient strains showing relatedness to multiple extant modern lineages. Denser temporal and spatial sampling of European *P. falciparum* may help to elucidate whether these strains did indeed result from multiple admixture events or constitute a deeply diverged population without closely related extant lineages. More broadly, sampling of additional ancient and archival materials provides an opportunity to generate a more-comprehensive catalogue of *Plasmodium* diversity. Such efforts may be especially beneficial for regional populations in which

successful elimination campaigns limit opportunities for sampling in public-health contexts. Similarly, although the near-fixation of the Duffy-negative allele limits *P. vivax* endemicity in sub-Saharan Africa today, ancient genome-wide data would provide an ideal opportunity to address debates regarding the geographic origins of this species. Despite preservation problems, our recovery of *P. vivax* DNA from ancient Egypt demonstrates that genotyping ancient *Plasmodium* strains from tropical and subtropical regions is theoretically possible. Finally, the ability to identify specific parasites in particular regions and time periods sets the stage for renewed study of the economic and human impact of malaria on past cultures. Integrating evidence from ancient DNA with historical records, osteological markers of anaemia and archaeological data could shed new light on historical debates, such as the possible role of malaria in the decline of the ancient Greek and/or Roman civilizations. Taken together, the capacity to reconstruct ancient genomes from *Plasmodium* spp. lays the groundwork for future studies on the origins, transmission, evolution and cultural impact of human malaria parasites.

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

Megan Michel[1,2,3✉], Eirini Skourtanioti[1,3], Federica Pierini[1], Evelyn K. Guevara[1,4], Angela Mötsch[1,3], Arthur Kocher[1,5], Rodrigo Barquera[1], Raffaela A. Bianco[1,3], Selina Carlhoff[1], Lorenza Coppola Bove[1,3,6], Suzanne Freilich[1,7], Karen Giffin[1,8], Taylor Hermes[1,9], Alina Hiß[1], Florian Knolle[10], Elizabeth A. Nelson[11], Gunnar U. Neumann[1,3], Luka Papac[1], Sandra Penske[1], Adam B. Rohrlach[1,12,13], Nada Salem[1,3], Lena Semerau[1], Vanessa Villalba-Mouco[1,3,14], Isabelle Abadie[15,16], Mark Aldenderfer[17], Jessica F. Beckett[18], Matthew Brown[19], Franco G. R. Campus[20], Tsang Chenghwa[21], María Cruz Berrocal[22], Ladislav Damašek[23], Kellie Sara Duffett Carlson[24], Raphaël Durand[25,26], Michal Ernée[27], Cristinel Fântăneanu[28], Hannah Frenzel[29], Gabriel García Atiénzar[30], Sonia Guillén[31], Ellen Hsieh[21], Maciej Karwowski[32], David Kelvin[33], Nikki Kelvin[34], Alexander Khokhlov[35], Rebecca L. Kinaston[36,37], Arkadii Korolev[35], Kim-Louise Krettek[38], Mario Küßner[39], Luca Lai[40], Cory Look[19], Kerttu Majander[41], Kirsten Mandl[7], Vittorio Mazzarello[42], Michael McCormick[3,43], Patxuka de Miguel Ibáñez[30,44,45], Reg Murphy[46], Rita E. Németh[47], Kerkko Nordqvist[48], Friederike Novotny[49], Martin Obenaus[50], Lauro Olmo-Enciso[51], Päivi Onkamo[52], Jörg Orschiedt[53,54], Valerii Patrushev[55], Sanni Peltola[1,56], Alejandro Romero[30,57], Salvatore Rubino[42], Antti Sajantila[4,58], Domingo C. Salazar-García[59,60], Elena Serrano[61,62], Shapulat Shaydullaev[63], Emanuela Sias[64], Mario Šlaus[65], Ladislav Stančo[23], Treena Swanston[66], Maria Teschler-Nicola[7,49], Frederique Valentin[67], Katrien Van de Vijver[68,69,70], Tamara L. Varney[71], Alfonso Vigil-Escalera Guirado[72], Christopher K. Waters[73], Estella Weiss-Krejci[74,75,76], Eduard Winter[49], Thiseas C. Lamnidis[1], Kay Prüfer[1], Kathrin Nägele[1], Maria Spyrou[1,77], Stephan Schiffels[1], Philipp W. Stockhammer[1,3,78], Wolfgang Haak[1], Cosimo Posth[1,38,77], Christina Warinner[1,3,79], Kirsten I. Bos[1], Alexander Herbig[1✉] & Johannes Krause[1,3✉]

[1]Department of Archaeogenetics, Max Planck Institute for Evolutionary Anthropology, Leipzig, Germany. [2]Department of Human Evolutionary Biology, Harvard University, Cambridge, MA, USA. [3]Max Planck-Harvard Research Center for the Archaeoscience of the Ancient Mediterranean, https://archaeoscience.org. [4]Department of Forensic Medicine, University of Helsinki, Helsinki, Finland. [5]Transmission, Infection, Diversification and Evolution Group, Max Planck Institute of Geoanthropology, Jena, Germany. [6]Department of Legal Medicine, Toxicology and Physical Anthropology, University of Granada, Granada, Spain. [7]Department of Evolutionary Anthropology, University of Vienna, Vienna, Austria. [8]Department of Environmental Sciences, University of Basel, Basel, Switzerland. [9]Department of Anthropology, University of Arkansas, Fayetteville, AR, USA. [10]Department of Medical Engineering and Biotechnology, University of Applied Sciences Jena, Jena, Germany. [11]Microbial Palaeogenomics Unit, Department of Genomes and Genetics, Institut Pasteur, Paris, France. [12]School of Computer and Mathematical Sciences, University of Adelaide, Adelaide, Australia. [13]Adelaide Data Science Centre, University of Adelaide, Adelaide, Australia. [14]Instituto Universitario de Investigación en Ciencias Ambientales de Aragón, IUCA-Aragosaurus, Universitity of Zaragoza, Zaragoza, Spain. [15]Inrap – Institut national de recherches archéologiques préventives, Paris, France. [16]Centre Michel de Boüard, Centre de recherches archéologiques et historiques anciennes et médiévales, Université de Caen Normandie, Caen, France. [17]Department of Anthropology and Heritage Studies, University of California, Merced, Merced, CA, USA. [18]Independent consultant, Cagliari, Sardinia, Italy. [19]Sociology and Anthropology Department, Farmingdale State College, Farmingdale, NY, USA. [20]Department of History, Human Sciences, and Education, University of Sassari, Sassari, Italy. [21]Institute of Anthropology, National Tsing Hua University, Hsinchu, Taiwan. [22]Institute of Heritage Sciences (INCIPIT), Spanish National Research Council (CSIC), Santiago de Compostela, Spain. [23]Institute of Classical Archaeology, Faculty of Arts, Charles University, Prague, Czech Republic. [24]Human Evolution and Archaeological Sciences, University of Vienna, Vienna, Austria. [25]Service d'archéologie préventive Bourges plus, Bourges, France. [26]UMR 5199 PACEA, Université de Bordeaux, Pessac Cedex, France. [27]Department of Prehistoric Archaeology, Institute of Archaeology of the Czech Academy of Sciences, Prague, Czech Republic. [28]National Museum of Unification Alba Iulia, Alba Iulia, Romania. [29]Anatomy Institute, University of Leipzig, Leipzig, Germany. [30]Instituto Universitario de Investigación en Arqueología y Patrimonio Histórico, Universidad de Alicante, San Vicente del Raspeig (Alicante), Spain. [31]Centro Mallqui, Lima, Peru. [32]Institut für Urgeschichte und Historische Archäologie, University of Vienna, Vienna, Austria. [33]Department of Microbiology and Immunology, Dalhousie University, Halifax, Nova Scotia, Canada. [34]Division of Ancient Pathogens, BioForge Canada Limited, Halifax, Nove Scotia, Canada. [35]Samara State University of Social Sciences and Education, Samara, Russia. [36]BioArch South, Waitati, New Zealand. [37]Griffith Centre for Social and Cultural Studies, Griffith University, Nathan, Queensland, Australia. [38]Senckenberg Centre for Human Evolution and Palaeoenvironment, University of Tübingen, Tübingen, Germany. [39]Thuringian State Office for Heritage Management and Archaeology, Weimar, Germany. [40]Department of Anthropology, University of North Carolina at Charlotte, Charlotte, NC, USA. [41]Department of Environmental Science, Integrative Prehistory and Archaeological Science, University of Basel, Basel, Switzerland.

[42]Department of Biomedical Sciences, University of Sassari, Sassari, Italy. [43]Initiative for the Science of the Human Past at Harvard, Department of History, Harvard University, Cambridge, MA, USA. [44]Servicio de Obstetricia, Hospital Virgen de los Lirios-Fisabio, Alcoi, Spain. [45]Sección de Antropología, Sociedad de Ciencias Aranzadi, Donostia - San Sebastián, Spain. [46]University of Nebraska-Lincoln, Lincoln, NE, USA. [47]Mureș County Museum, Târgu Mureş, Romania. [48]Helsinki Collegium for Advanced Studies, University of Helsinki, Helsinki, Finland. [49]Department of Anthropology, Natural History Museum Vienna, Vienna, Austria. [50]Silva Nortica Archäologische Dienstleistungen, Thunau am Kamp, Austria. [51]Department of History, University of Alcalá, Alcalá de Henares, Spain. [52]Department of Biology, University of Turku, Turku, Finland. [53]Landesamt für Denkmalpflege und Archäologie Sachsen-Anhalt, Halle, Germany. [54]Institut für Prähistorische Archäologie, Freie Universität Berlin, Berlin, Germany. [55]Centre of Archaeological and Ethnographical Investigation,  Mari State University, Yoshkar-Ola, Russia. [56]Faculty of Biological and Environmental Sciences, University of Helsinki, Helsinki, Finland. [57]Departamento de Biotecnología, Universidad de Alicante, San Vicente del Raspeig, Spain. [58]Forensic Medicine Unit, Finnish Institute for Health and Welfare, Helsinki, Finland. [59]Departament de Prehistòria, Arqueologia i Història Antiga, Universitat de València, Valencia, Spain. [60]Department of Geological Sciences, University of Cape Town, Cape Town, South Africa. [61]Instituto Internacional de Investigaciones Prehistóricas, Universidad de Cantabria, Santander, Spain. [62]TAR Arqueología, Madrid, Spain. [63]Faculty of History, Termez State University, Termez, Uzbekistan. [64]Centro Studi sulla Civiltà del Mare, Stintino, Italy. [65]Anthropological Center, Croatian Academy of Sciences and Arts, Zagreb, Croatia. [66]Department of Anthropology, Economics and Political Science, MacEwan University, Edmonton, Alberta, Canada. [67]UMR 8068, CNRS, Nanterre, France. [68]Royal Belgian Institute of Natural Sciences, Brussels, Belgium. [69]Center for Archaeological Sciences, University of Leuven, Leuven, Belgium. [70]Dienst Archeologie - Stad Mechelen, Mechelen, Belgium. [71]Department of Anthropology, Lakehead University, Thunder Bay, Ontario, Canada. [72]Departamento de Humanidades: Historia, Geografía y Arte, Universidad Carlos III de Madrid, Getafe, Spain. [73]Heritage Department, National Parks of Antigua and Barbuda, St. Paul's Parish, Antigua and Barbuda. [74]Austrian Archaeological Institute, Austrian Academy of Sciences, Vienna, Austria. [75]Institut für Ur- und Frühgeschichte, Heidelberg University, Heidelberg, Germany. [76]Department of Social and Cultural Anthropology, University of Vienna, Vienna, Austria. [77]Archaeo- and Palaeogenetics, Institute for Archaeological Sciences, Department of Geosciences, University of Tübingen, Tübingen, Germany. [78]Institute for Pre- and Protohistoric Archaeology and Archaeology of the Roman Provinces, Ludwig Maximilian University, Munich, Germany. [79]Department of Anthropology, Harvard University, Cambridge, MA, USA. [✉]e-mail: megan_michel@eva.mpg.de; alexander_herbig@eva.mpg.de; krause@eva.mpg.de

## Methods

### Samples

All individuals analysed were sampled as part of previous studies following legal regulations and ethical guidelines specific to the region of origin in each case. We confirm that appropriate permissions were obtained to perform the analyses described.

### *Plasmodium* screening

To identify candidates for *Plasmodium* capture, we used the Heuristic Operations for Pathogen Screening pipeline to screen more than 10,000 shotgun-sequencing datasets previously produced by the Max Planck Institute (MPI) of Geoanthropology (formerly the MPI for the Science of Human History) and/or the MPI for Evolutionary Anthropology against a custom database containing *Plasmodium* species of interest as well as potential contaminant taxa[63] (Supplementary Methods 1 and Supplementary Tables 9 and 10). To reduce false-positive species assignment caused by cross-mapping, ancient shotgun data were pre-processed to exclude reads aligning to the human genome from subsequent analysis. Adapter trimming and merging was performed using leeHom (v.1.1.5-eb382b3 or v.1.1.5-ba378b6) using the flag --ancientdna, and reads were mapped using BWA aln (v.0.7.12) with the following parameters: -n 0.01, -o 2 and -l 16500 (refs. 64,65). Samtools (v.1.3) was used for indexing and filtering of unmapped sequences, and low-complexity reads were removed using a parallelized implementation of PRINSEQ (-lc_method dust, -lc_threshold 7; https://github.com/spabinger/prinseq_parallel; refs. 66,67). Read alignment and taxonomic binning were performed using the MetaGenome Analyzer (MEGAN) Alignment Tool (MALT, v.0.5.2), which was executed with the following parameters: BlastN mode with semiglobal alignment, minimum support value for the LCA algorithm of 1 (-sup 1), maximum alignments per query of 100 (-mq 100), top percent value for the LCA algorithm of 1 (-top 1) and minimum percent identity used by the LCA algorithm (-mpi) set to 90 (ref. 68). We assessed edit distance and damage rates of reads assigned to *Plasmodium* spp. using the Heuristic Operations for Pathogen Screening pipeline[63]. After this screening, a large proportion of libraries had one or more reads assigned to *P. falciparum*, *P. vivax* and/or *P. malariae*. However, visual analysis of candidate alignments using MEGAN confirmed that many reads exhibited high edit distances when aligned to *Plasmodium* spp., stacking in presumably conserved regions and/or low sequence complexity[69]. To further reduce our candidate list, we required at least one assigned read aligning with no mismatches to a *Plasmodium* species of interest, and reads were further evaluated using the Basic Local Alignment Search Tool (BLAST) web interface to assess sequence specificity (https://blast.ncbi.nlm.nih.gov/Blast.cgi).

### Hybridization-capture design

To increase the percentage of *Plasmodium* DNA in our ancient libraries, we generated two new in-solution DNA bait sets targeting the *Plasmodium* mitochondrial and nuclear genomes. Baits for each genomic compartment were designed and implemented separately owing to difficulties controlling for variation in copy number between nuclear and organellar genomes[70]. For each probe set, we compiled a list of target *Plasmodium* spp. and downloaded relevant reference sequences from the National Center for Biotechnology Information (NCBI) (Supplementary Table 11). Only assembled chromosomes were used in the nuclear-capture design. After reference selection, genomes for each capture array were combined and low-complexity regions were masked using dustmasker with default parameters (v.1.0.0, from the BLAST package 2.9.0)[71]. Concatenated sequences were used to generate 52-base pair (bp) probes with an 8-bp linker sequence; we used tiling densities of 1 bp and 6 bp for the mitochondrial and nuclear probe sets, respectively[72]. Finally, probe sequences were subjected to complexity filtering and duplicate removal using a custom script (ProbeGenerator v.0.89, written by A.H.), resulting in 32,634 and 2,897,533 probes for the mitochondrial and nuclear captures, respectively. After several quality checks (Supplementary Methods 2), we ordered 1 million feature Agilent SureSelect DNA capture arrays for both the mitochondrial ($n = 1$) and nuclear ($n = 3$) probe sets, replicating probes to fully utilize space. Baits were cleaved from the array surfaces to generate two in-solution hybridization-capture reagents, as described elsewhere[72].

### Laboratory processing

After metagenomic screening, we identified 36 ancient individuals from 26 archaeological sites exhibiting evidence of *Plasmodium* DNA preservation (Supplementary Table 1). At least one skeletal element from each individual had previously been sampled for ancient DNA in dedicated clean-room facilities at the MPI of Geoanthropology (formerly the MPI for the Science of Human History) in Jena, Germany, the Institute for Archaeological Sciences in Tübingen, Germany, and/or the Laboratories of Molecular Anthropology and Microbiome Research at the University of Oklahoma in Norman (Supplementary Table 12). Most of the samples were obtained from teeth ($n = 27$), although we also analysed 7 samples from the petrous portion of the temporal bone and 2 calcified tissue specimens (Supplementary Table 12). For the individual LDC020 from the site of Laguna de los Cóndores in Peru, two separate teeth were sampled for subsequent analyses.

For tooth samples, powdered dentine was obtained from the interior of the pulp chamber (protocol at https://doi.org/10.17504/protocols.io.bqebmtan). Samples processed in Jena and Tübingen were subjected to various decontamination strategies, including ultraviolet irradiation, sandblasting and/or bleach treatment, after which the teeth were sectioned at the enamel–dentine junction and powder was obtained from the dental pulp chamber by drilling. Sampling of CHO001 was performed at the University of Oklahoma as described[50]. Petrous bones were processed as previously described with minor modifications (protocol at https://doi.org/10.17504/protocols.io.bqd8ms9w). COR001 was sectioned longitudinally before drilling from the cut face, and TAQ018 was processed using a sandblaster before sampling from the otic capsule[73]. Finally, two specimens (GAT004 and TOR008) consisting of calcified tissue were sampled using customized approaches. For GAT004, a small piece of calcified material was removed, washed in EDTA for 15 min to reduce contamination and then incubated overnight in standard extraction buffer (0.45 M EDTA, 0.25 mg ml$^{-1}$ proteinase K, pH 8). TOR008 was sampled by drilling from the outside of the calcified nodule.

In total, 40 DNA extracts were generated from the 37 skeletal samples described above (Supplementary Table 12). For 26 samples from 25 unique individuals, DNA was extracted in a dedicated clean-room facility at either the MPI of Geoanthropology or the Institute for Archaeological Sciences using around 40–100 mg powdered dentine and following a modified version of a published silica-column-based protocol[74] (https://doi.org/10.17504/protocols.io.baksicwe). For 11 samples, the decalcification and denaturation step of the extraction was performed at the MPI of Geoanthropology, after which the lysate was collected, frozen and shipped to the MPI for Evolutionary Anthropology in Leipzig, Germany, for further processing. Finally, for the remaining three samples, lysate preparation was performed in a dedicated clean-room facility at the MPI for Evolutionary Anthropology as previously described, except that 0.05% Tween-20 was added to the extraction buffer[74,75]. Using an automated liquid-handling system (Bravo NGS Workstation B, Agilent Technologies), DNA was purified from 150 µl lysate using silica-coated magnetic beads and binding buffer D with a final elution volume of 30 µl (ref. 75). Extraction blanks without sample material were carried alongside the samples during DNA extraction.

Next, 45 ancient DNA libraries were generated following single- and/or double-stranded library preparation protocols optimized for the recovery of ancient DNA (Supplementary Table 12). Three non-UDG-treated and five UDG-half-treated double-stranded DNA libraries were prepared at the MPI of Geoanthropology or the Institute for Archaeological Sciences following previously published protocols[76,77]

(https://doi.org/10.17504/protocols.io.bmh6k39e and https://doi.org/10.17504/protocols.io.bakricv6). All double-stranded indexed libraries were purified, quantified using quantitative PCR and amplified using IS5/IS6 primers and a Herculase II fusion DNA polymerase (Agilent) to a final concentration of around 200–400 ng μl⁻¹. A further 35 DNA libraries were prepared from 30 μl extract using an automated version of the previously described single-stranded partial UDG DNA library preparation protocol[78,79]. *E. coli* uracil-DNA-glycosylase and *E. coli* endonuclease VIII were added to the dephosphorylation master mix during library preparation to remove uracils found inside molecules. Library yields and efficiency of library preparation were determined using two quantitative PCR assays[79]. The remaining two libraries were prepared from 30 μl extract using an automated version of the single-stranded DNA library preparation protocol without UDG treatment. All libraries were amplified and tagged with pairs of sample-specific indices using AccuPrime *Pfx* DNA polymerase, and amplified libraries were purified using SPRI technology[79,80]. Libraries were prepared from both the sample DNA extracts and the extraction blanks, and further negative controls (library blanks) were added. Finally, all sample and control libraries were enriched for *Plasmodium* spp. mitochondrial and/or nuclear DNA using two consecutive rounds of in-solution hybridization capture performed on the Bravo NGS workstation B[72]. Captured libraries were sequenced either on a HiSeq4000 with 75 bp single-end sequencing chemistry (1 × 76 + 8 + 8 cycles) or on a NextSeq500 with 75 bp paired-end sequencing chemistry (2 × 76 + 8 + 8 cycles). A further 41 libraries from the sites of St. Rombout's cemetery in Mechelen and Laguna de los Cóndores were captured using the 1,240k SNP capture array[72,81,82], which is widely used in genome-wide studies of human ancient DNA (Supplementary Methods 3).

## Data pre-processing

Sequencing data from mitochondrial and nuclear-capture experiments were preprocessed and analysed using nf-core/eager (v.2.4.5) with double- and single-stranded libraries treated separately owing to differences in the reverse adapter sequences[83]. AdapterRemoval (v.2.3.2) was used for trimming and, where necessary, read merging[84]. Ambiguous and low-quality bases (quality threshold below 20) were removed while preserving 5′ read ends (--preserve5p). We retained reads of 30 bp and above and required a minimum adapter overlap of 1 bp for trimming. For libraries sequenced on the NextSeq500, polyg trimming with fastp (v.0.20.1) was enabled using the nf-core/eager flag --complexity_filter_poly_g; we also trimmed 2 bp from the 5′ and 3′ ends of reads from UDG-half-treated libraries[85]. Lanes were merged and the quality of sequencing data was evaluated with FastQC (https://www.bioinformatics.babraham.ac.uk/projects/fastqc/). Finally, we used the tool AMDirT to identify and download publicly available ancient *Plasmodium* spp. shotgun-sequencing datasets from the Sequence Read Archive (ERR3649966, ERR3649967, ERR3650017, ERR3650065, ERR3650068, ERR3650072 and ERR3651363)[30,86]. These pre-processed sequences derive from several blood slides produced in Spain's Ebro Delta around 1944; previous analyses found that the slides contain DNA from both *P. falciparum* and *P. vivax*[30,34,35]. To identify the species present, we competitively mapped our pre-processed capture data to concatenated references including either mitochondrial or nuclear genome sequences from target human-infecting *Plasmodium* spp. (Supplementary Methods 4 and 5).

## Analysis of *Plasmodium* mitochondrial genomes

After capture quality control (Supplementary Methods 4), we mapped pre-processed reads from *P. falciparum*-, *P. vivax*- and *P. malariae*-positive libraries to the appropriate mitochondrial reference (LR605957.1, LT635627.1 and LT594637.1, respectively). For co-infected individuals, we included only reads competitively mapped to the respective species to avoid false-positive SNP calls. Mapping was performed with BWA aln (v.0.7.17-r1188) using loose (-n 0.01, -l 16) and strict

(-n 0.1, -l 32) mapping parameters for non-UDG and UDG-half libraries, respectively[64]. Alignments were filtered using Samtools (v.1.12) with a mapping quality threshold of 37, and duplicates were removed using Picard MarkDuplicates with default parameters (http://broadinstitute.github.io/picard/)[66]. Alignments from multiple libraries were merged on an individual level using nf-core/eager (v.2.4.6), and bam files were processed using Picard AddOrReplaceReadGroups for downstream compatibility[83]. Genotyping was performed with the GATK UnifiedGenotyper (v.3.5) with default parameters and using the output mode 'EMIT_ALL_SITES'[87]. Combining ancient genotypes with publicly available modern mitochondrial data (Supplementary Table 13 and Supplementary Methods 6), we generated an SNP alignment using MultiVCFAnalyzer (v.0.85.2), requiring a minimal coverage of 3× for a base call and excluding positions from problematic regions[88] (Supplementary Methods 4). For single-stranded non-UDG-treated libraries (Supplementary Table 12), we used GenoSL to perform a damage-aware genotyping, considering potential damaged bases on the forward- and reverse-mapping reads separately[89] (https://github.com/aidaanva/GenoSL). Using a custom script, we excluded genomes with less than 90% genome coverage and performed complete deletion filtering with the tool MDF.R (https://github.com/aidaanva/MDF). After removing any invariant sites, we output the curated SNP alignment and regional population assignments in NEXUS format. To visualize patterns of relatedness for the mitochondrial genomes, we constructed median-joining networks using PopART (http://popart.otago.ac.nz).

## *Plasmodium* nuclear genotyping

For libraries that passed nuclear-capture quality control (Supplementary Methods 5), we extracted alignments competitively mapped to the *P. vivax* and/or *P. falciparum* nuclear chromosome scaffolds and converted them to FASTQ format using BEDtools[90] (v.2.25.0). We rationalized that this pre-filtering step would reduce the number of potential false-positive SNP calls arising from cross-species mapping in multi-species co-infections. Positive capture libraries were mapped to the 14 nuclear chromosome scaffolds of the *P. falciparum* Pf3D7 (refs. 91,92) (GCA_000002765.3) and/or the *P. vivax* PvP01 (ref. 93) (GCA_900093555.1) reference(s); as for the mitochondrial-capture data, damage-trimmed single- and double-stranded UDG-half libraries were mapped using BWA aln (v.0.7.17) with strict parameters (-n 0.1, -l 32), and non-UDG-treated libraries were mapped using loose parameters (-n 0.01, -l 16)[64]. After using Samtools (v.1.12) for alignment filtering (-q 37), we removed duplicates using Picard MarkDuplicates with default parameters[66] (http://broadinstitute.github.io/picard/). To prevent genotyping errors arising from ancient-DNA damage in non-UDG-treated libraries, we used the trimBam utility from BamUtil (v.1.0.15) to clip 7 bp from the 5′ and 3′ ends of aligned reads[94]. De-duplicated, pre-processed bams from the same individual were merged and haploid genotypes were called using the nf-core/eager implementation of pileupcaller (v.1.5.2, https://github.com/stschiff/sequenceTools) with default parameters (--run_genotyping, --genotyping_tool pileupcaller). For *P. falciparum*- and *P. vivax*-positive samples, we called genotypes at 873,060 and 872,564 high-quality SNP positions ascertained in modern datasets released as part of the MalariaGEN *P. falciparum* Community Project and the MalariaGEN *P. vivax* Genome Variation Project, respectively[37,38] (Supplementary Methods 7). After pruning of the modern datasets (Supplementary Methods 7), we removed positions at which the minor allele was present in fewer than two strains, resulting in 106,179 and 419,387 sites segregating in the modern *P. falciparum* and *P. vivax* populations, respectively. Ancient and modern datasets were merged with genotypes from appropriate outgroups (Supplementary Methods 8) for subsequent population genetic analysis.

## *Plasmodium* population genetic analysis

For *P. vivax* and *P. falciparum* population genetic analysis, we merged data from 906 modern and 23 ancient strains and 1,227 modern

and 9 ancient strains, respectively (Supplementary Methods 7). For each species, we qualitatively assessed the genetic affinity between modern and ancient strains using smartPCA (v.16000, lsqproject YES, shrinkmode YES, outliermode 2, usenorm NO), computing eigenvectors based on high-coverage modern genotypes and projecting ancient data onto these axes of variation[95,96]. To identify appropriate SNP coverage cutoffs for our ancient datasets, we downsampled modern strains to simulate low-coverage ancient samples and evaluated their performance in PCA (Supplementary Methods 9). On the basis of these experiments, we considered 5,000 segregating SNPs to be an appropriate threshold for reliably differentiating strains from distinct populations. Although we included ancient strains with coverage as low as 500 segregating SNPs in our PCA analyses, we point out that precise positioning in PCA space should be interpreted with great caution at this level of coverage.

Next, we used unsupervised ADMIXTURE (v.1.3.0) to assess the population structure of modern *P. falciparum* and *P. vivax* using a model-based approach[97]. After converting genotype data to binary format, we used PLINK (v.1.90; http://pngu.mgh.harvard.edu/purcell/plink/) to filter variants with a minor allele frequency below 1% (--make-bed, --maf) and prune variants with a correlation threshold above 0.4 using a 200-bp sliding window and a step size of 25 (--indep-pairwise 200 25 0.4)[98]. In total, we retained 108,438 and 20,107 SNPs for *P. vivax* and *P. falciparum*, respectively. For each species, we performed five replicate runs of unsupervised ADMIXTURE for each value of $K$ between 2 and 15. We compiled ADMIXTURE output and visualized results using open-source scripts (written by T.C.L.; https://github.com/TCLamnidis/AdmixturePlotter). After evaluating the CV Errors to determine the best value of $K$ for each species, we performed supervised ADMIXTURE to model the ancient *P. falciparum* and *P. vivax* strains as mixtures of the modern populations maximizing each component[97]. For *P. vivax*, we used the following six populations as sources: Eastern Southeast Asia, Ethiopia, Latin America, Oceania, Western Asia and Western Southeast Asia. For *P. falciparum*, we modelled our ancient strains using the following eight modern populations: East Africa, West Africa, South America, South Asia, Western Southeast Asia, Eastern Southeast Asia, Indonesia and Papua New Guinea. For each species, we performed 200 bootstrap replicates to assess the reliability of ADMIXTURE modelling on our low-coverage ancient datasets (-B). After observing that ancient European *P. falciparum* strains are modelled as complex mixtures of multiple populations using this approach, we repeated the supervised ADMIXTURE analysis using Ebro1944 as an extra source population.

For both *P. falciparum* and *P. vivax*, we used qp3Pop and qpDstat from the AdmixTools (v.7.0.2) suite to quantitatively test specific hypotheses about the relatedness of modern and ancient *Plasmodium* populations[99]. For both species, we evaluated outgroup $F_3$ statistics of the form $f_3$(Test1, Test2; Outgroup), where Test1 and Test2 include all modern populations and high-coverage ancient strains. For *P. falciparum*, we used West African populations as an outgroup to explore affinities between non-African modern and ancient populations. For *P. vivax*, we selected *P. vivax*-like as an outgroup to increase SNP counts and improve our statistical power to differentiate population relatedness. To test for cladality of ancient *P. vivax* strains with modern Latin American (LAM) populations, we ran $F_4$ statistics of the form $f_4$(*P. vivax*-like, Ancient; Test, LAM), with Test referring to modern populations other than LAM. To test for cladality between modern African and South American (SAM) *P. falciparum*, we computed $F_4$ statistics of the form $f_4$(*P. praefalciparum*, Test; SAM, Africa), where Africa includes the West African (WAF), East African (EAF) and Central African (CAF) populations and Test includes both non-African modern populations and high-coverage ancient strains.

Next, we used MEGA-CC (v.10.0.2) to construct neighbour-joining trees for modern and ancient *P. falciparum* and *P. vivax*, respectively[100]. For each species, we excluded ancient strains with fewer than 5,000 segregating SNPs genotyped. We used *P. vivax*-like and *P. praefalciparum* as outgroups for the *P. vivax* and *P. falciparum* analyses, respectively. After converting our eigenstrat-format SNP data to multifasta alignments (EigenToFasta.py, https://github.com/meganemichel/plasmodium_project_scripts), we built neighbour-joining trees using the following parameters: pairwise deletion mode, 100 bootstrap replicates, Jukes Cantor substitution model, and rate variation following a gamma distribution with parameter 1.00. The resulting phylogenies were visualized using the Interactive Tree of Life (v.6.7.6)[101].

## Human population genetic analysis

For a subset of ancient individuals from malaria-positive sites, we performed human population genetic analysis as described in Supplementary Methods 10.

## Reporting summary

Further information on research design is available in the Nature Portfolio Reporting Summary linked to this article.

## Data availability

Raw sequencing data from 36 malaria-positive individuals, as well as newly reported data from 41 ancient individuals enriched at human ancestry-informative SNP positions, have been deposited at the European Nucleotide Archive (accession number PRJEB73276). Ancient and modern *P. vivax* and *P. falciparum* nuclear genotypes are available in eigenstrat format (https://figshare.com/projects/Ancient_Plasmodium_genomes_shed_light_on_the_history_of_human_malaria/196711). This study used modern *P. falciparum* and *P. vivax* genotype datasets available through the Pf6 data release of the MalariaGEN *P. falciparum* Community Project (ftp://ngs.sanger.ac.uk/production/malaria/pfcommunityproject/Pf6/) and the Pv4 data release of the MalariaGEN *P. vivax* Genome Variation project (ftp://ngs.sanger.ac.uk/production/malaria/Resource/30). Previously published raw sequencing datasets from Indian *P. falciparum* strains and the Ebro Delta blood slide can be obtained from the European Nucleotide Archive under accession numbers PRJNA322219 and PRJEB30878, respectively. This study used previously published ancient human genotype datasets obtained from the Reich laboratory's Allen Ancient DNA Resource v.54.1 (https://reich.hms.harvard.edu/allen-ancient-dna-resource-aadr-downloadable-genotypes-present-day-and-ancient-dna-data). Previously published modern *P. falciparum* and *P. vivax* mitochondrial datasets, as well as genomic sequences used in our probe design and metagenomic screening database, are available via the National Center for Biotechnology Information (accession numbers can be found in Supplementary Tables 9, 10, 11 and 13). The following whole-genome sequencing datasets obtained from the NCBI Sequence Read Archive were used for phylogenetic dating: SAMN02677154, SAMN02677164, SAMN02677167, SAMN03274512, SAMN02677169, SAMN02677170, SAMN02677171, SAMN02677180, SAMN02677183, SAMN02677184, SAMN02677185, SAMN02677186, SAMN02677187, SAMN02677195 and SAMN00710542. The following genome assemblies and chromosome sequences available via the NCBI were used as references in this study: *P. falciparum* mitochondria: LR605957.1; *P. vivax* mitochondria: LT635627.1; *P. malariae* mitochondria: LT594637.1; *P. falciparum* nuclear chromosomes: GCA_000002765.3; *P. vivax* nuclear chromosomes: GCA_900093555.1; *P. vivax*-like nuclear chromosomes: GCA_003402215.1; *Plasmodium cynomolgi* nuclear chromosomes: GCA_900180395.1; *P. praefalciparum* nuclear chromosomes: GCA_900095595.1; and the Genome Reference Consortium Human Build 37 (GRCh37): PRJNA31257. Maps presented in the main text and Extended Data figures were produced using the following resources: Cartopy (v.0.20.3, https://github.com/SciTools/cartopy/tree/v0.20.3); Natural Earth (naturalearthdata.com); and World Shaded Relief map (Esri, 2009).

## Code availability

Custom scripts used for data processing and/or analysis can be retrieved from https://github.com/meganemichel/plasmodium_project_scripts.

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

**Acknowledgements** This project was funded by the National Science Foundation, grants BCS-2141896 and BCS-1528698; the European Research Council (ERC) under the European Union's Horizon 2020 programme, grants 851511-MICROSCOPE (to S. Schiffels), 771234-PALEoRIDER (to W.H.) and starting grant 805268-CoDisEASe (to K.I.B.); and the ERC starting grant Waves ERC758967 (supporting K. Nägele and S.C.). We thank the Max Planck-Harvard Research Center for the Archaeoscience of the Ancient Mediterranean for supporting M. Michel, E. Skourtanioti, A.M., R.A.B., L.C.B., G.U.N., N.S., V.V.-M., M. McCormick, P.W.S., C.W. and J.K.; the Kone Foundation for supporting E.K.G. and A.S.; and the Faculty of Medicine and the Faculty of Biological and Environmental Sciences at the University of Helsinki for grants to E.K.G. A.S. thanks the Magnus Ehrnrooth Foundation, the Sigrid Jusélius Foundation, the Finnish Cultural Foundation, the Academy of Finland, the Life and Health Medical Foundation and the Finnish Society of Sciences and Letters. M.C.B. acknowledges funding from: research project PID2020-116196GB-I00 funded by MCIN/AEI/10.13039/501100011033; the Spanish Ministry of Culture; the Chiang Ching Kuo Foundation; Fundación Palarq; the EU FP7 Marie Curie Zukunftskolleg Incoming Fellowship Programme, University of Konstanz (grant 291784); STAR2-Santander Universidades and Ministry of Education, Culture and Sports; and CEI 2015 project Cantabria Campus Internacional. M.E. received support from the Czech Academy of Sciences award Praemium Academiae and project RVO 67985912 of the Institute of Archaeology of the Czech Academy of Sciences, Prague. This work has been funded within project PID2020-115956GB-I00 'Origen y conformación del Bronce Valenciano', granted by the Ministry of Science and Innovation of the Government of Spain, and grants from the Canadian Institutes for Health Research (MZI187236), Research Nova Scotia (RNS 2023-2565) and The Center for Health Research in Developing Countries. D.K. is the Canada research chair in translational vaccinology and inflammation. R.L.K. acknowledges support from a 2019 University of Otago research grant (Human health and adaptation along Silk Roads, a bioarchaeological investigation of a medieval Uzbek cemetery). P.O. thanks the Jane and Aatos Erkko Foundation, the Finnish Cultural Foundation and the Academy of Finland. S. Peltola received support from the Emil Aaltonen Foundation and the Ella and Georg Ehrnrooth Foundation. D.C.S.-G. thanks the Generalitat Valenciana (CIDEGENT/2019/061). E.W.K. acknowledges support from the DEEPDEAD project, HERA-UP, CRP (15.055) and the Horizon 2020 programme (grant 649307). M. Spyrou thanks the Elite program for postdocs of the Baden-Württemberg Stiftung. We thank all the institutions and individuals who facilitated access to the samples analysed in this study. We thank the Peruvian Ministry of Culture, Centro Mallqui, and Museo Leymebamba for access to the samples from Laguna de los Cóndores. Some of the data was produced by the Ancient DNA Core Unit of the Max Planck Institute for Evolutionary Anthropology, which is funded by the Max Planck Society. We thank E. Essel for coordinating sample processing by the core unit; L. Jáuregui for information about laboratory protocols; B. Nickel for preparing *Plasmodium* capture probes; the staff of the archaeogenetics laboratory at the Max Planck Institute for Evolutionary Anthropology for performing sampling and for dsDNA library production and capture; H. Gruber for helping us identity anomalous burial contexts; K. McMullen for copy-editing the manuscript; and E. Carrión Santafé, M. Contreras Martínez, E. Baquedano Perez and I. Shirobokov for helping us to access archaeological remains. This publication uses data from the MalariaGEN *P. falciparum* Community Project as described in ref. 38 and the MalariaGEN *P. vivax* Genome Variation Project as described in ref. 37.

**Author contributions** A. Herbig and J.K. initiated and supervised the study. S.F., E.K.G., A.M., K. Nägele, M. Spyrou, S. Schiffels, P.W.S., W.H., C.P., C.W., K.I.B., A. Herbig, and J.K. coordinated the collection of samples for genetic analysis; I.A., M.A., J.F.B., M.B., F.G.R.C., T.C., M.C.B., L.D., K.S.D.C., R.D., M.E., C.F., H.F., G.G.A., S.G., E.H., M. Karwowski, D.K., N.K., A. Khokhlov, R.L.K., A. Korolev, K.-L.K., M. Küßner, L.L., C.L., K. Majander, K. Mandl, V.M., M. McCormick, P.d.M.I., R.M., R.E.N., K. Nordqvist, F.N., M.O., L.O.-E., P.O., J.O., V. P., S. Peltola, A.R., S.R., A.S., D.C.S.-G., E. Serrano, S. Shaydullaev, E. Sias, M. Šlaus, L. Stančo, T.S., M.T.-N., F.V., K.v.d.V., T.L.V., A.V.-E.G., C.K.W., E.W.-K., and E.W. participated in archaeological excavation, provided anthropological assessment and/or contributed to the curation of skeletal materials. R.B., R.A.B., S.C., L.C.B., S.F., K.G., E.K.G., T.H., A. Hiß, F.K., E.A.N., G.U.N., L.P., S. Penske, A.B.R., N.S., L. Semerau, and V.V.-M. managed samples and metadata and/or conducted laboratory work. M. Michel, T.C.L., and K.P. performed bioinformatic analysis. M. Michel conducted metagenomic screening of ancient sequencing datasets. M. Michel and A. Herbig designed and evaluated *Plasmodium* mitochondrial and nuclear probe sets. M. Michel curated the ancient malaria datasets and performed computational analyses. E. Skourtanioti supported the human and pathogen population genetic analyses. F.P., E.K.G. and E. Skourtanioti analysed the ancient human genome-wide data. A.M. compiled archaeological metadata and edited site descriptions. A. Kocher statistically analysed the effect of skeletal-element type on *Plasmodium* DNA preservation in archaeological materials. M. Michel, A. Herbig and J.K. drafted the manuscript with contributions from all co-authors.

**Funding** Open access funding provided by Max Planck Society.

**Competing interests** The authors declare no competing interests.

**Additional information**
**Correspondence and requests for materials** should be addressed to Megan Michel, Alexander Herbig or Johannes Krause.

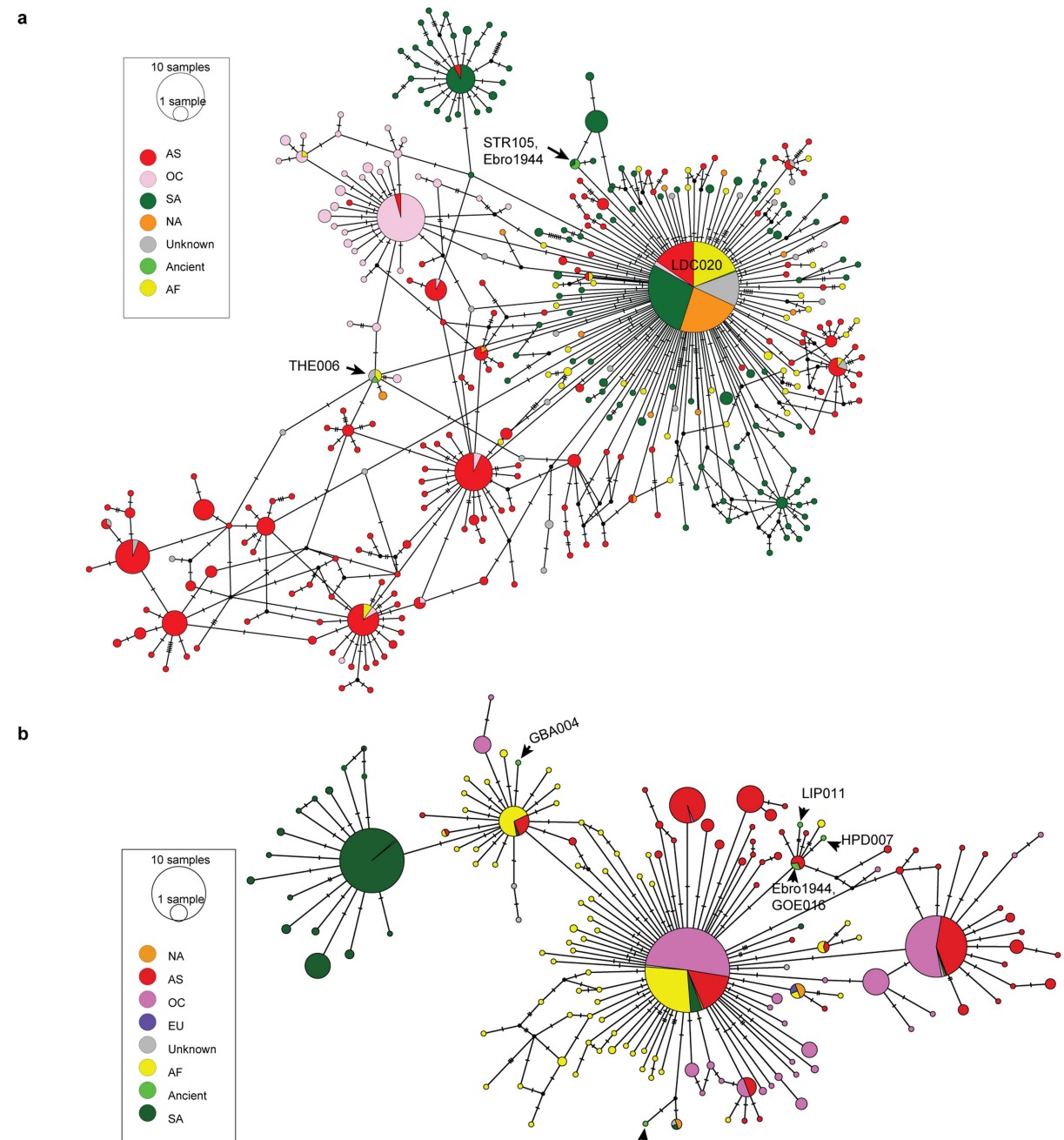

**Extended Data Fig. 1 | *P. falciparum* and *P. vivax* mitochondrial networks.**
**a**. Median-joining network showing relatedness between modern *P. vivax* mitochondrial haplotypes and ancient *P. vivax* strains. **b**. Median-joining network showing relatedness between modern *P. falciparum* mitochondrial haplotypes and ancient *P. falciparum* strains. In both a and b, circles represent haplotypes with size proportional to the number of sampled strains. Hatch marks indicate the number of mutational steps separating haplotypes, while color reflects the geographic origin of modern strains.

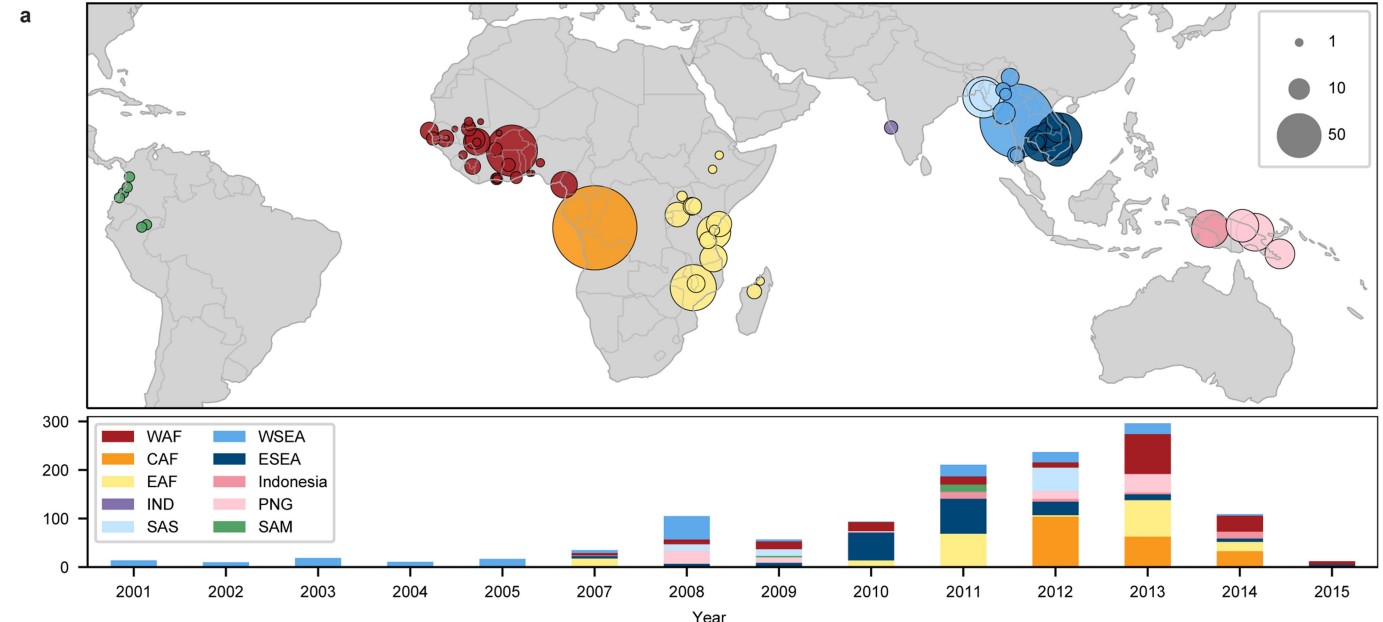

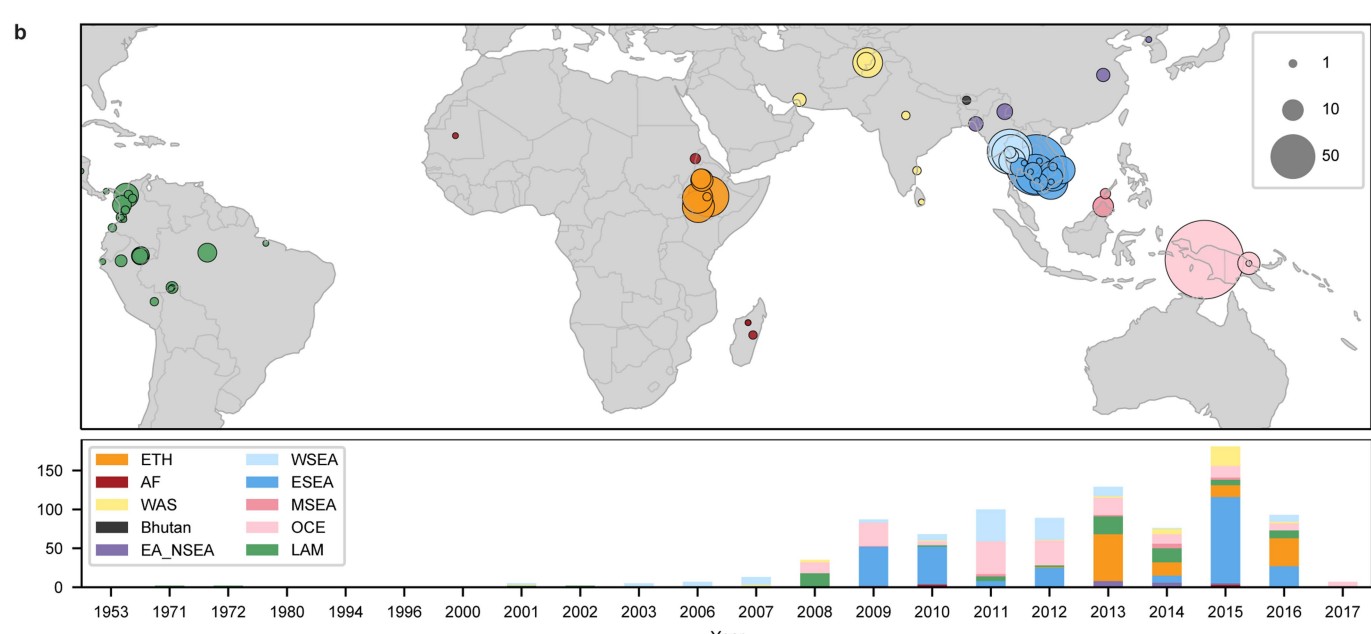

**Extended Data Fig. 2 | Modern *P. falciparum* and *P. vivax* datasets.**
**a**. Geographic origins, population assignments, and sampling years for 1,232
*P. falciparum* strains included in our modern comparative dataset. Genotypes
and metadata derive from the Pf6 release of the MalariaGEN *P. falciparum*
Community Project (n = 1,227) as well as previously published next-generation
sequencing datasets from India (n = 5)[38,45]. **b**. Geographic origins, population
assignments, and sampling years for 1,055 *P. vivax* strains included in our

modern comparative dataset. Metadata and genotypes were published as part
of the Pv4 release of the MalariaGEN *P. vivax* Genome Variation Project[37]. For
most samples, population assignments mirror those from the original data
release; however, strains annotated as 'unassigned' have been given population
labels based both on genetic analysis and their region of origin. Both maps were
produced using Cartopy (v. 0.20.3, https://github.com/SciTools/cartopy/tree/
v0.20.3) with Natural Earth (naturalearthdata.com).

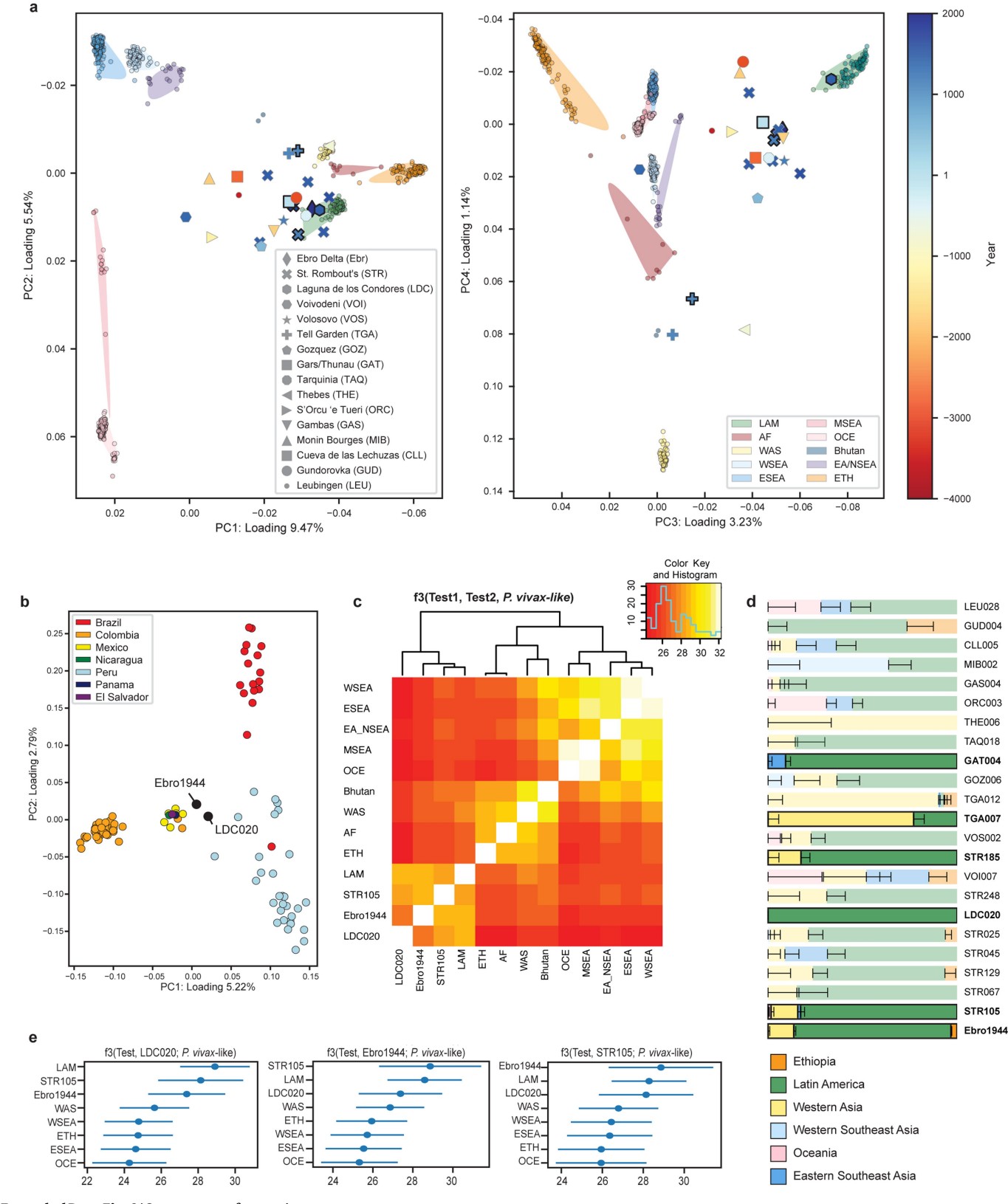

**Extended Data Fig. 3** | See next page for caption.

**Extended Data Fig. 3 | *P. vivax* population genetic analysis. a**. PCA showing all ancient *P. vivax* samples. Symbols corresponding to high-coverage strains (> 5,000 segregating SNPs covered) are outlined with a thick stroke. Marker shape corresponds to the archaeological site and color reflects the mean age of *P. vivax* strains from that site. Ancient clones are projected onto the diversity of modern *P. vivax* strains (small circles). Shaded regions delimit the spread of modern *P. vivax* populations in PCA space. **b**. PCA showing LDC020 and Ebro1944 projected onto the diversity of modern Latin American *P. vivax* strains. **c**. $F_3$ statistics showing shared drift between pairs of modern *P. vivax* populations and ancient strains relative to the outgroup *P. vivax*-like. Lighter colors reflect higher affinity between populations. **d**. Supervised ADMIXTURE analysis modeling all ancient *P. vivax* strains as mixtures of the following K = 6 source populations: Ethiopia (ETH), Latin America (LAM), Western Asia (WAS), Western Southeast Asia (WSEA), Oceania (OCE), and Eastern Southeast Asia (ESEA). Error bars represent standard errors for the mean admixture proportions estimated using 300 bootstrap replicates. Transparent bars indicate low-coverage *P. vivax* samples (< 5,000 segregating SNPs covered). **e**. Selected $F_3$ statistics showing affinity between the ancient European and Latin American *P. vivax* strains (LDC020, Ebro1944, and STR105) and the following modern *P. vivax* populations: LAM (n = 96), WAS (n = 44), WSEA (n = 126), ETH (n = 129), ESEA (n = 277), and OCE (n = 189). Error bars represent ± 3 standard errors.

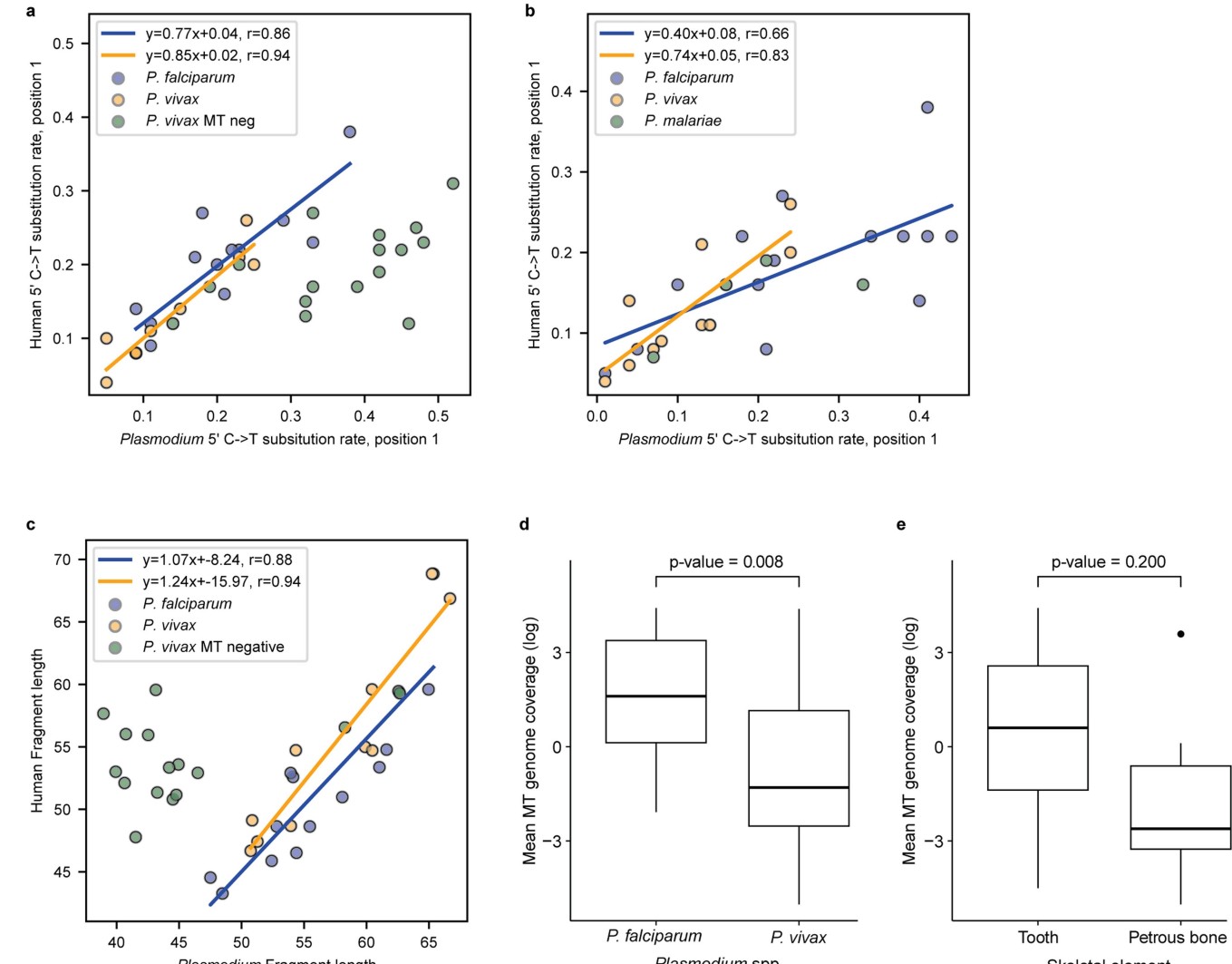

**Extended Data Fig. 4 | Preservation Characteristics. a**. For *Plasmodium* nuclear-captured libraries, comparison of 5' C to T substitution rates on position 1 of reads mapping to target *Plasmodium* spp. and the human genome. *P. vivax* MT negative refers to samples that failed to yield MT capture data passing quality control metrics (Methods). **b**. Per-library comparison of 5' C to T substitution rates on position 1 of reads mapping to target *Plasmodium* spp. and the human genome following mitochondrial capture. **c**. Comparison of mean fragment lengths of reads mapping to target *Plasmodium* spp. and the human reference genome following nuclear capture. Linear regressions were computed using the SciPy stats package. Panels d-e show boxplots of the mean *Plasmodium* mitochondrial genome coverage obtained from each aDNA library depending on **d**. the *Plasmodium* species present (*P. vivax*: n = 23, *P. falciparum*: n = 14) and **b**. the skeletal element from which DNA was extracted (teeth: n = 30, pars petrosa: n = 7). The reported p-values were estimated using a linear mixed model accounting for the non-independence of libraries prepared from the same individuals. Each box shows the lower quartile, median, and upper quartile of the dataset. Whiskers delimit the range of observations between the upper or lower quartile and 1.5 times the interquartile range.

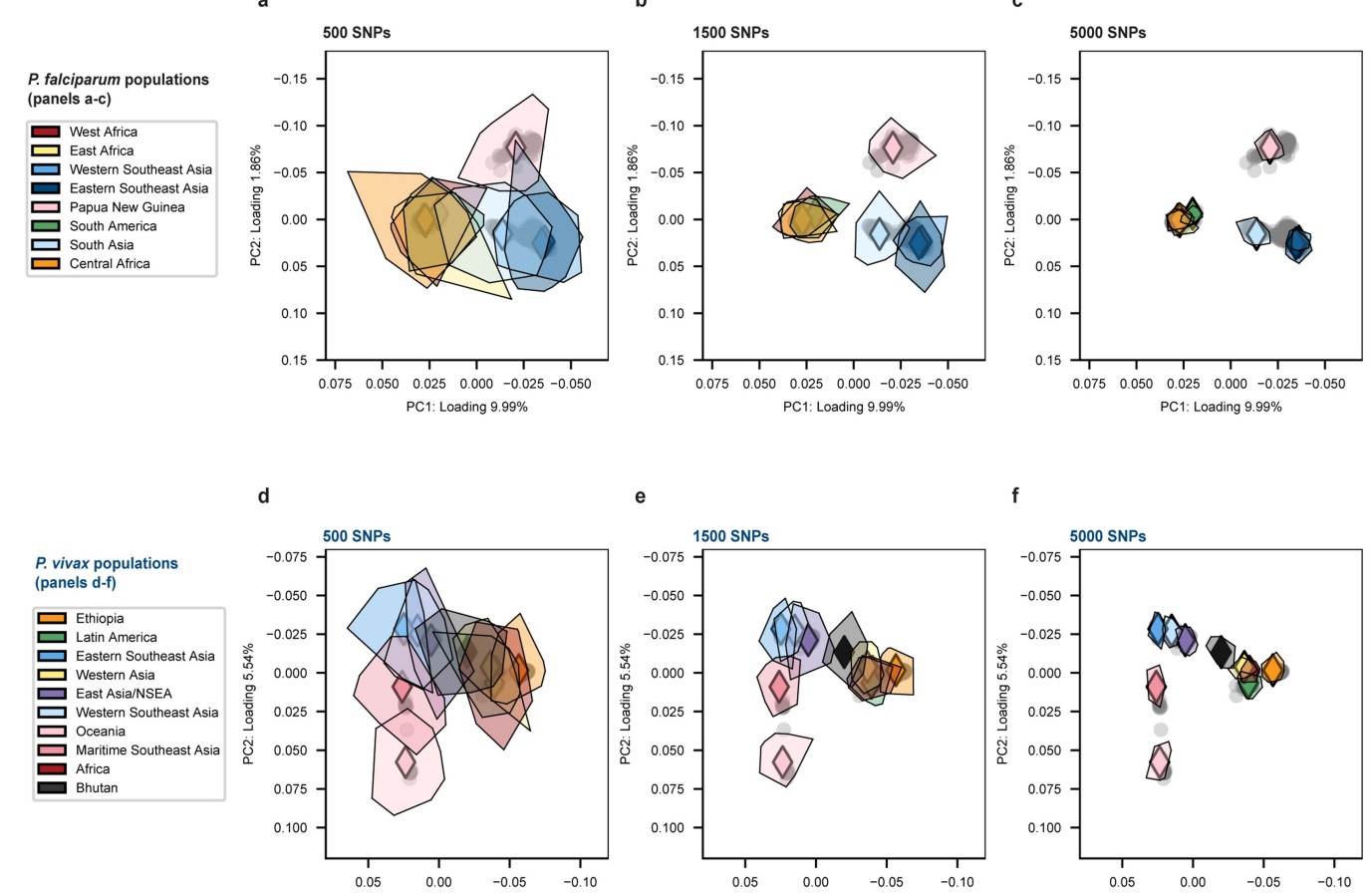

**Extended Data Fig. 5 | *P. falciparum* and *P. vivax* coverage simulations.** Principal components analysis showing impact of SNP coverage on population ascertainment in ancient *P. falciparum* (a-c) and ancient *P. vivax* (d-f) datasets. Diamonds reflect the PCA position of one randomly selected modern strain from each population, which was downsampled to the following coverage levels: **a**. and **d**. 500 segregating SNPs, **b**. and **e**. 1,500 segregating SNPs, and **c**. and **f**. 5,000 segregating SNPs. Downsampled strains were projected onto modern clones using smartPCA. Colored polygons delineate the spatial distribution of 50 downsampled replicates for each population/coverage combination.

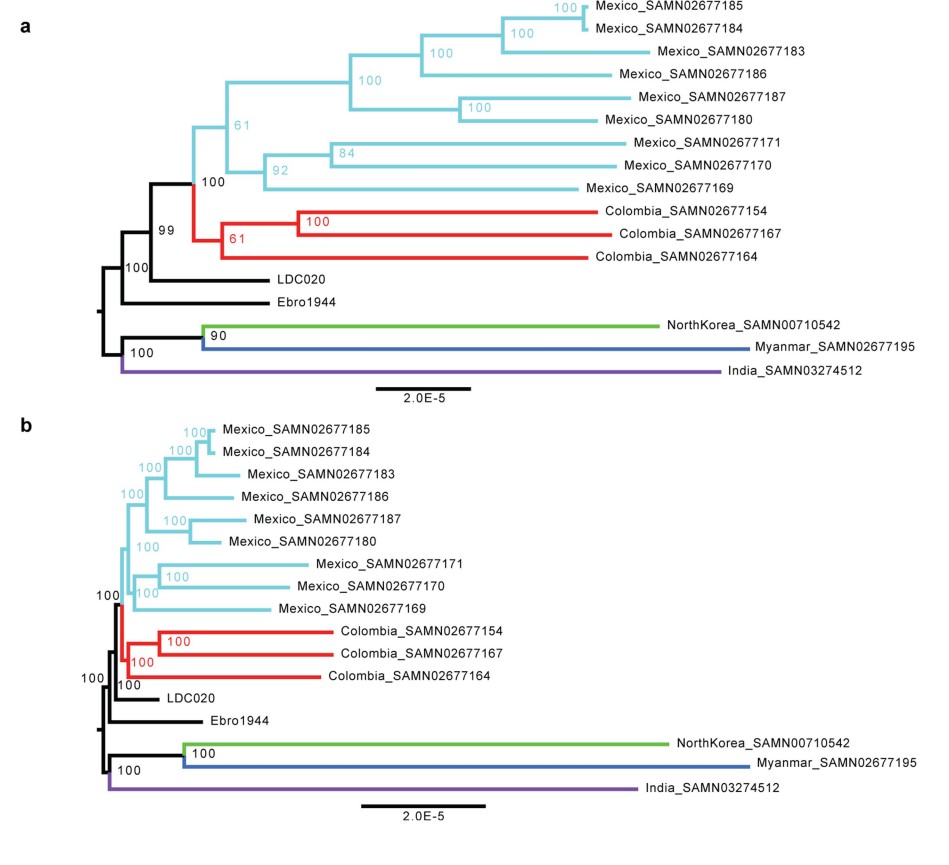

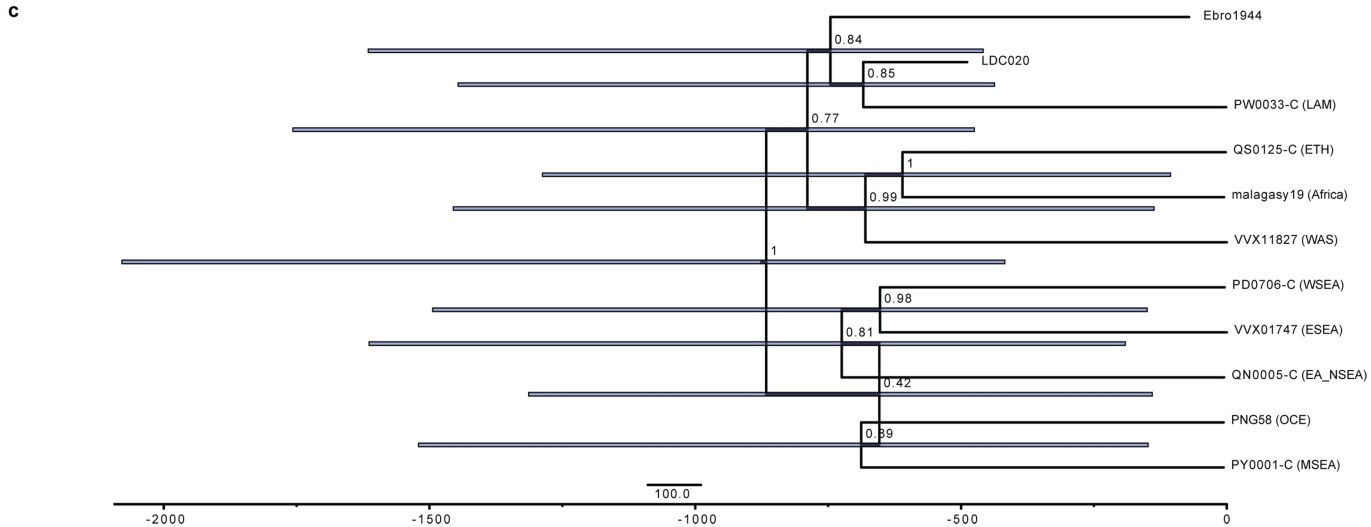

**Extended Data Fig. 6 | Bayesian Molecular Dating Using BEAST.** a. Maximum likelihood phylogeny produced with RAxML-NG using a complete deletion alignment including 17,100 SNP positions. Numbers reflect support values estimated using 1,000 bootstrap replicates. b. Maximum likelihood phylogeny produced with RAxML-NG using a homoplasy-stripped complete deletion alignment including 11,977 SNP positions. Numbers reflect support values estimated using 1,000 bootstrap replicates. c. Maximum clade credibility tree produced using BEAST2 with the optimized relaxed clock and Bayesian coalescent skyline models. Bars represent median node heights and the x-axis reflects years before the present. Numbers represent node posterior probabilities. Leaves are annotated with both sample name and population labels.

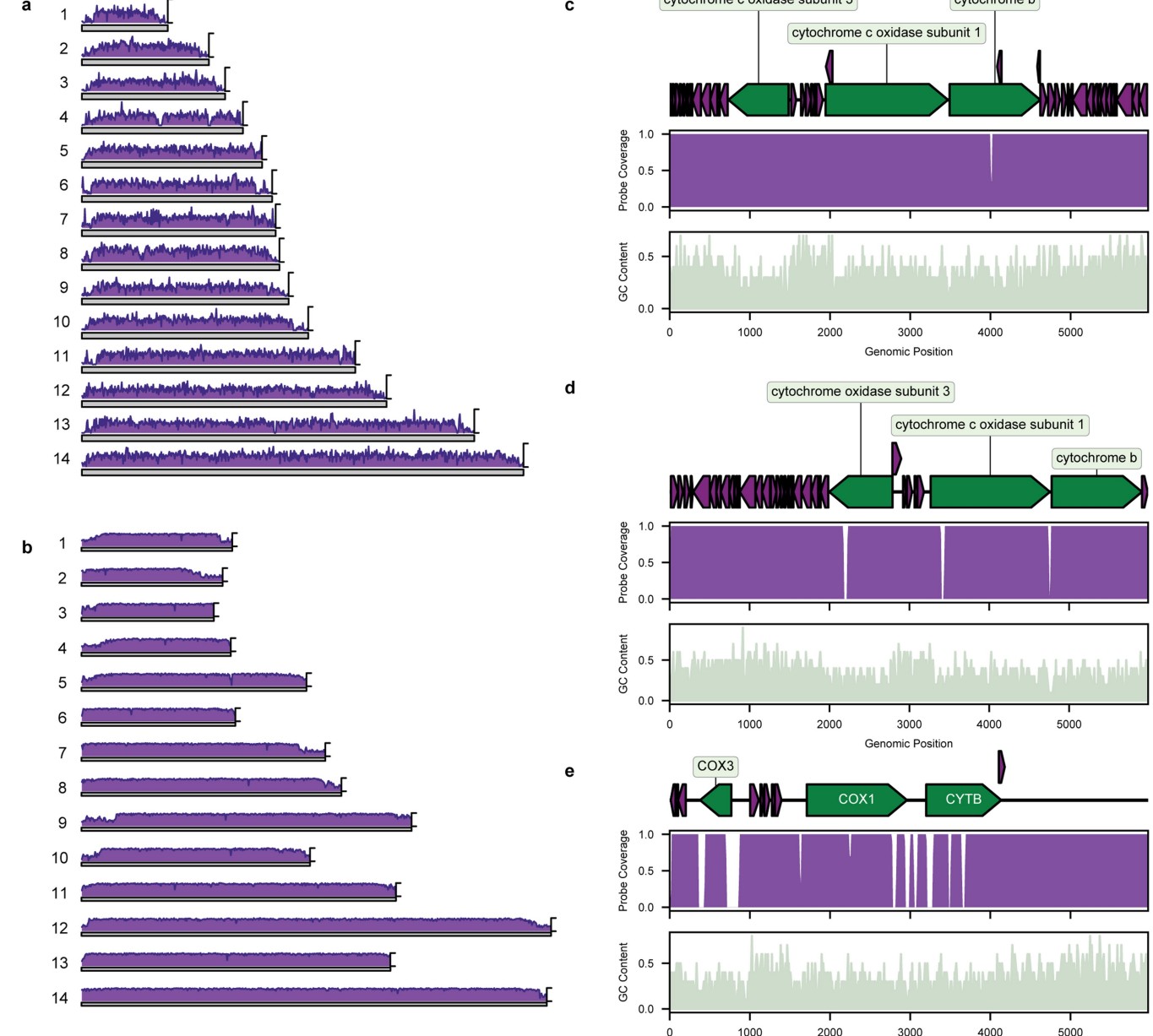

**Extended Data Fig. 7 | Mitochondrial and nuclear capture coverage.** Panels a-b show mapping of the nuclear probes to the following reference genomes: **a**. *P. falciparum* 3D7 (GCA_000002765.3) and **b**. *P. vivax* PvP01 (GCA_900093555.1). For each species, the 14 nuclear chromosomes are shown, with the purple track indicating the proportion of each 5,000 bp window covered by at least one probe. The nuclear probes span 99.8% and 53.6% of the *P. vivax* and *P. falciparum* references, respectively, which is consistent with the higher prevalence of low-complexity AT-rich regions in the *P. falciparum* reference genome. Panels **c**–**e** show mapping of the mitochondrial probes to the following mitochondrial references: **c**. *P. falciparum* (LR605957.1), **d**. *P. vivax* (LT635627.1), **e**. *P. malariae* (LT594637.1). For each species, the top track shows the protein coding sequences and rRNA content in green and purple, respectively. The middle track shows the fraction of each 10 bp window covered by at least one probe, and the bottom track shows GC content within each 10 bp window.

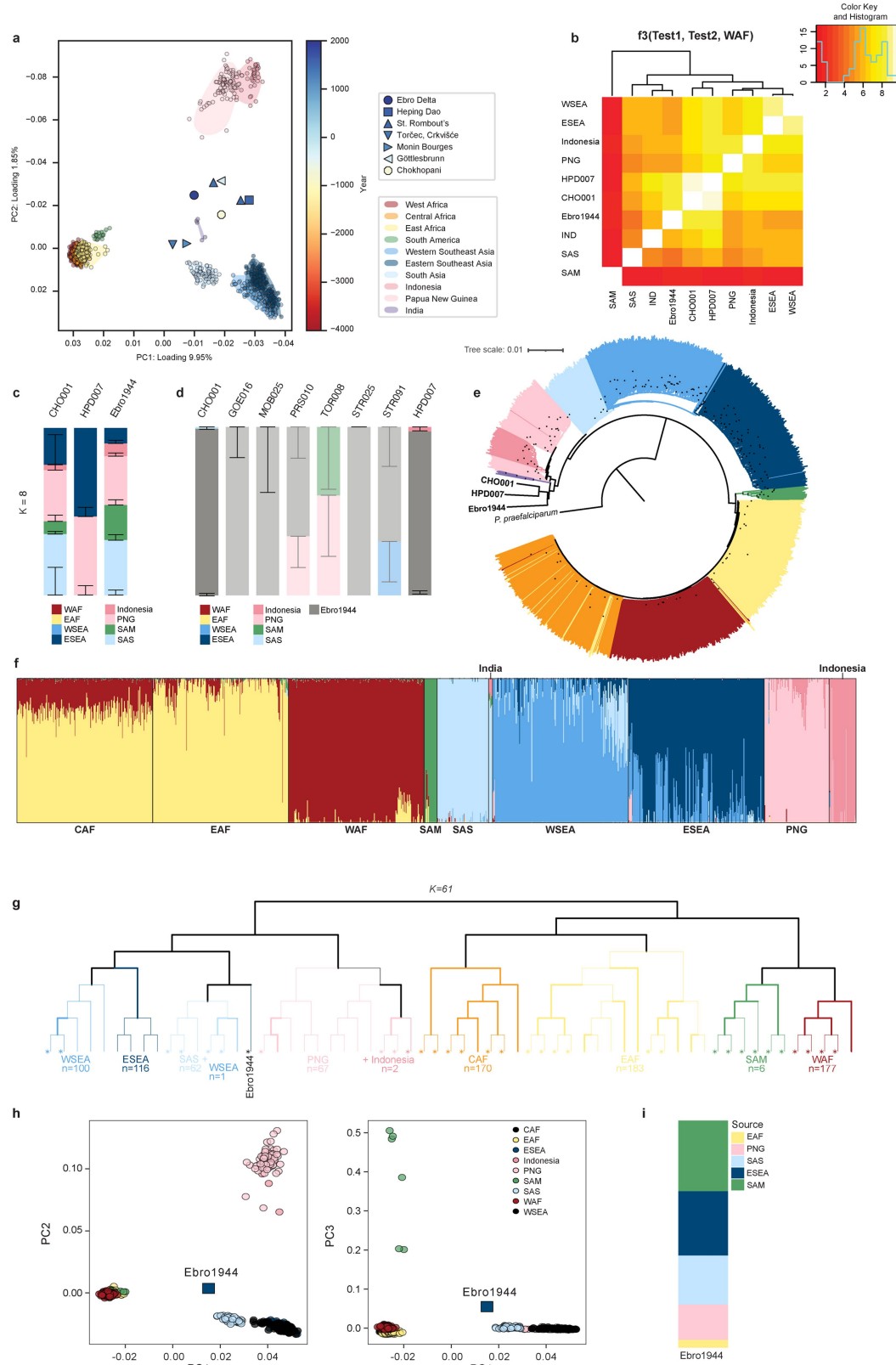

**Extended Data Fig. 8** | See next page for caption.

**Extended Data Fig. 8 | *P. falciparum* population genetic analysis. a**. PCA showing all ancient *P. falciparum* strains, including low coverage samples (<10,000 segregating SNPs covered). Marker shape corresponds to the archaeological site, and color reflects the mean age of *P. falciparum* strains per site. Ancient data is projected onto the diversity of modern *P. falciparum* strains (small circles). Shaded regions show the spread of modern *P. falciparum* populations in PCA space. **b**. F3-statistics showing shared drift between pairs of modern *P. falciparum* populations and ancient strains relative to the outgroup WAF. Lighter colors reflect higher affinity between populations. **c**. Supervised analysis modeling mean ADMIXTURE proportions for high coverage ancient *P. falciparum* strains (>10,000 SNPs) using the following K = 8 source populations: West Africa (WAF), East Africa (EAF), Western Southeast Asia (WSEA), Eastern Southeast Asia (ESEA), Indonesia, Papua New Guinea (PNG), South America (SAM), and South Asia (SAS). Error bars give standard errors estimated using 300 bootstrap replicates. **d**. Supervised ADMIXTURE analysis modeling all ancient *P. falciparum* strains as mixtures of K = 9 source populations, including both those enumerated in panel c and the ancient genome Ebro1944. Error bars give standard errors of the mean admixture proportions estimated using 300 bootstrap replicates. Transparent bars indicate low-coverage *P. falciparum* samples (< 10,000 segregating SNPs covered). **e**. Neighbor-joining phylogeny including ancient and modern *P. falciparum* strains. Branches are colored by strain geographic origin as in panel a. Black points reflect nodes receiving support values of greater than or equal to 0.9 (100 bootstrap replicates). **f**. Unsupervised ADMIXTURE analysis of modern *P. falciparum* strains, with components colored according to the modern population in which that source is maximized. **g**. Tree of genetic relationships among the 61 populations inferred by the unlinked model of fineSTRUCTURE. Thick stroke indicates sample groupings receiving ≥ 90% posterior probability after 500 samples of the MCMC chain. Terminal branches representing a single strain are indicated with an asterisk. **h**. PCA computed on the output coancestry matrix places Ebro1944 closer to *P. falciparum* strains from South Asia. **i**. The output of Chromopainter analysis was analyzed with GLOBETROTTER to determine which populations, corresponding to broader geographical regions, are required to model Ebro1944.

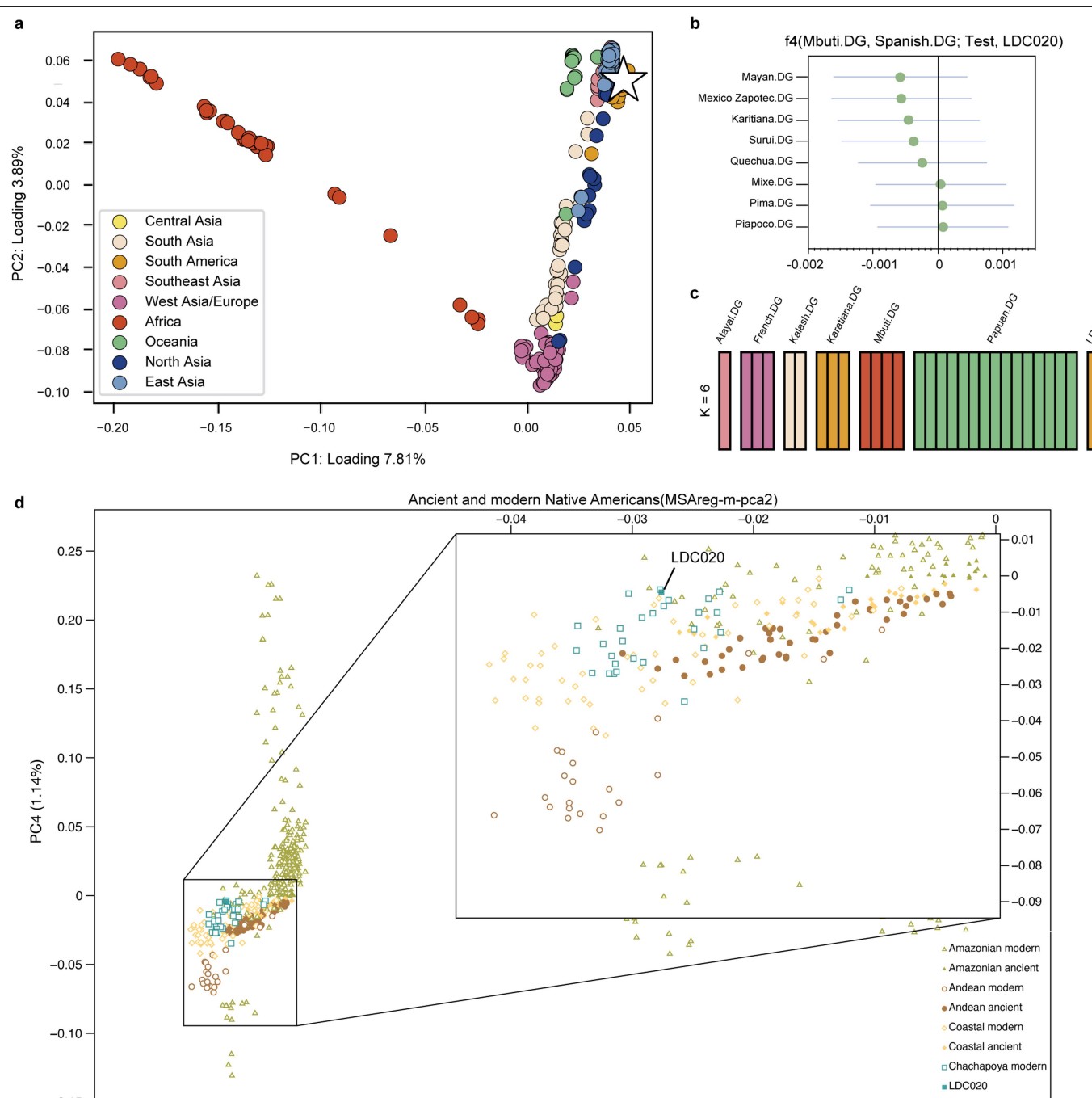

**Extended Data Fig. 9 | LDC020 Human Population Genetic Analysis. a**. PCA computed using a global set of modern human populations. LDC020 was projected onto these axes of variation (white star). **b**. $F_4$-statistics testing for cladality between LDC020 and a test panel of the following modern South American Indigenous populations with negligible European admixture: Mayan (n = 2), Mexico Zapotec (n = 2), Karitiana (n = 3), Surui (n = 2), Quechua (n = 3), Mixe (n = 3), Pima (n = 2), and Piapoco (n = 2). A negative statistic indicates excess allele sharing between either Spanish.DG (n = 2) and the test population or LDC020 and Mbuti.DG (n = 4), while a positive statistic supports excess affinity between either Spanish.DG and LDC020 or Mbuti.DG and the test population. Error bars show the point estimate for the $F_4$ statistic ±3 standard errors. **c**. Supervised ADMIXTURE analysis modeling the ancestry of LDC020 as a composite of six modern populations: Atayal, French, Kalash, Karatiana, Mbuti, and Papuan. **d**. Regional PCA computed using selected ancient and modern populations from South America (Supplementary Methods 10). LDC020 is projected onto these axes of variation.

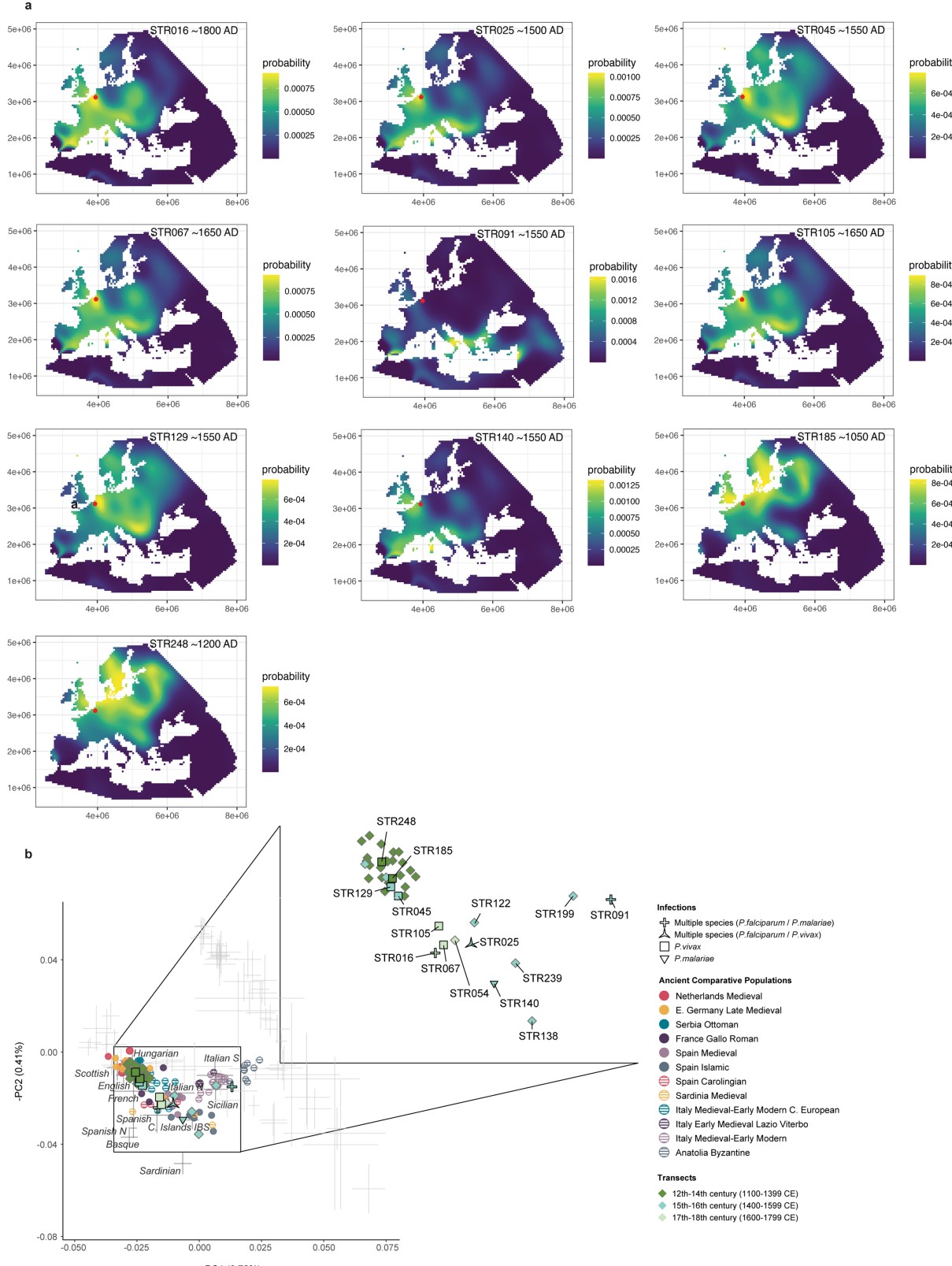

**Extended Data Fig. 10 | Human population genetics of malaria-positive individuals from Mechelen. a**. *Mobest* similarity probability search results for each of the ten malaria-infected individuals. The search was conducted in a range of ±100 years from the designated mean date of each individual. **b**. PCA including both malaria-infected and uninfected individuals from Mechelen projected onto the diversity of modern Western Eurasian populations. Marker type indicates infection status, while points are colored by time transect. Select ancient populations are visualized as colored circles for comparative purposes. Inset shows all individuals from Mechelen with the malaria-positives marked by individual labels.

# Reporting Summary

## Statistics

For all statistical analyses, confirm that the following items are present in the figure legend, table legend, main text, or Methods section.

| n/a | Confirmed | |
|---|---|---|
| ☐ | ☒ | The exact sample size (*n*) for each experimental group/condition, given as a discrete number and unit of measurement |
| ☒ | ☐ | A statement on whether measurements were taken from distinct samples or whether the same sample was measured repeatedly |
| ☐ | ☒ | The statistical test(s) used AND whether they are one- or two-sided<br>*Only common tests should be described solely by name; describe more complex techniques in the Methods section.* |
| ☒ | ☐ | A description of all covariates tested |
| ☒ | ☐ | A description of any assumptions or corrections, such as tests of normality and adjustment for multiple comparisons |
| ☐ | ☒ | A full description of the statistical parameters including central tendency (e.g. means) or other basic estimates (e.g. regression coefficient) AND variation (e.g. standard deviation) or associated estimates of uncertainty (e.g. confidence intervals) |
| ☐ | ☒ | For null hypothesis testing, the test statistic (e.g. *F*, *t*, *r*) with confidence intervals, effect sizes, degrees of freedom and *P* value noted<br>*Give P values as exact values whenever suitable.* |
| ☐ | ☒ | For Bayesian analysis, information on the choice of priors and Markov chain Monte Carlo settings |
| ☒ | ☐ | For hierarchical and complex designs, identification of the appropriate level for tests and full reporting of outcomes |
| ☒ | ☐ | Estimates of effect sizes (e.g. Cohen's *d*, Pearson's *r*), indicating how they were calculated |

*Our web collection on statistics for biologists contains articles on many of the points above.*

## Software and code

Policy information about availability of computer code

| | |
|---|---|
| Data collection | No software was used in data collection. |
| Data analysis | The following pieces of published/publicly available software were used in the analysis of data presented in this manuscript: AdapterRemoval (v. 2.3.2), AdmixTools (v. 7.0.2), ADMIXTURE (v. 1.3.0), AdmixturePlotter (https://github.com/TCLamnidis/AdmixturePlotter), AMDirT (v. 1.3, https://amdirt.readthedocs.io/en/latest/), ANGSD (v. 0.935), BamUtil (v. 1.0.15), Bayesian Evolutionary Analysis Sampling Trees 2 (BEAST2, v. 2.7.6), BEAUti (v. 2.7.6), BEDtools (v. 2.25.0 or v. 2.30.0), BLAST (https://blast.ncbi.nlm.nih.gov/Blast.cgi), Biopython (Bio.Entrez package, Biopython version 1.79), BREAD (), BWA aln (v. 0.7.12 or v. 0.7.17), ContamMix (v. 1.0-10), Chromopainter/fineSTRUCTURE (v. 2), DamageProfiler (v. 0.4.9 or v. 1.1), dustmasker (v. 1.0.0, from BLAST package 2.9.0), EIGENSOFT (v. 7.2.1), fastp (v. 0.20.1), FASTQC (v. 0.11.9, https://www.bioinformatics.babraham.ac.uk/projects/fastqc/), FigTree (v. 1.4.4), GATK UnifiedGenotyper (v. 3.5), GenoSL (https://github.com/aidaanva/GenoSL), Haplogrep3 (v. 3.2.1), Heuristic Operations for Pathogen Screening (HOPS, v. 0.35) pipeline, homoplasyFinder (https://github.com/JosephCrispell/homoplasyFinder), Interactive Tree of Live (iTOL, v. 6.7.6), ivar (v. 1.3), LcMLkin (v. ), leeHom (v. 1.1.5-eb382b3 or v. 1.1.5-ba378b6), lme4 (v. 1.1-34), MarkDuplicates (v. 2.26.0, http://broadinstitute.github.io/picard), MDF.R (https://github.com/aidaanva/MDF), MEGA-CC (v. 10.0.2), MEGAN Alignment Tool (MALT, v. 0.4.0, v. 0.4.1, or v. 0.5.2), MEGAN (MEtaGenome ANalyzer, v. 6.25.3), mobest (v. 1.0.0, https://github.com/nevrome/mobest/releases), MultiQC (v. 1.3), MultiVCFAnalyzer (v. 0.85.2 and v. 0.87.1), nf-core/eager (v. 2.3.1, v. 2.4.5, or v. 2.4.6), OxCal (v. 4.4), picard AddOrReplaceReadGroups (v. 2.18.29-SNAPSHOT, http://broadinstitute.github.io/picard), pileupcaller (v. 1.5.2), PLINK (v. 1.90, http://pngu.mgh.harvard.edu/purcell/plink/), pMMRCalculator (v. 1.1.0, https://github.com/TCLamnidis/pMMRCalculator), PMR (), PopART (http://popart.otago.ac.nz), Poseidon (v. 2.7.1, http://www.poseidon-adna.org), PRINSEQ-lite (v. 0.20.4), PRINSEQ parallel (https://github.com/spabinger/prinseq_parallel), ProbeGenerator (v. 0.89), qpAdm (v. )5.1, qpWave (v. 5.1), Qualimap (v. 2.2.2), randomise_dates_beast2.py (https://github.com/sebastianduchene/phylo_xml_tools), RAxML-NG (v. 1.1), READ (v. ), Samtools (v. 1.3, v. 1.9, or v. 1.12), SciPy (v. 1.9), SeqKit (v. 2.4.0), seqtk (v. 1.2-r94), Sex.DetERRmine (v 1.1.2), smartPCA (v. 16000), SNP-sites (v. 2.5.1), TempEst (v. 1.5.3), Tracer (v. 1.7.2), TreeAnnotator (v. 2.7.6). |

Custom scripts used for data processing and/or analysis can be retrieved from https://github.com/meganemichel/
plasmodium_project_scripts.

For manuscripts utilizing custom algorithms or software that are central to the research but not yet described in published literature, software must be made available to editors and reviewers. We strongly encourage code deposition in a community repository (e.g. GitHub). See the Nature Portfolio guidelines for submitting code & software for further information.

## Data

Policy information about availability of data

All manuscripts must include a data availability statement. This statement should provide the following information, where applicable:
- Accession codes, unique identifiers, or web links for publicly available datasets
- A description of any restrictions on data availability
- For clinical datasets or third party data, please ensure that the statement adheres to our policy

Raw sequencing data from 36 malaria-positive individuals as well as newly-reported data from 41 ancient individuals enriched at human ancestry-informative SNP positions have been deposited on the European Nucleotide Archive (ENA) (Accession number PRJEB73276). Ancient and modern P. vivax and P. falciparum nuclear genotypes are available in eigenstrat format (https://figshare.com/projects/ Ancient_Plasmodium_genomes_shed_light_on_the_history_of_human_malaria/196711). This study utilizes modern P. falciparum genotype datasets available through the Pf6 data release of the MalariaGEN P. falciparum Community Project (ftp://ngs.sanger.ac.uk/production/malaria/pfcommunityproject/Pf6/). Modern P. vivax genotype datasets analyzed here are available through the Pv4 data release of the MalariaGEN P. vivax Genome Variation project (ftp://ngs.sanger.ac.uk/ production/malaria/Resource/30). Previously published raw sequencing datasets from Indian P. falciparum strains and the Ebro Delta blood slide can be obtained from the European Nucleotide Archive under accession numbers PRJNA322219 and PRJEB30878, respectively. This study utilized previously published ancient human genotype datasets obtained from the Reich Laboratory's Allen Ancient DNA Resource v. 54.1 (https://reich.hms.harvard.edu/allen-ancient-dna-resource-aadr-downloadable-genotypes-present-day-and-ancient-dna-data). Previously published modern P. falciparum and P. vivax mitochondrial datasets as well as genomic sequences utilized in our probe design and metagenomic screening database are available from the National Center for Biotechnology Information (NCBI) (accession numbers in Supplementary Tables 9, 10, 11, and 13). The following whole-genome sequencing datasets available on the NCBI Sequence Read Archive were used for phylogenetic dating: SAMN02677154, SAMN02677164, SAMN02677167, SAMN03274512, SAMN02677169, SAMN02677170, SAMN02677171, SAMN02677180, SAMN02677183, SAMN02677184, SAMN02677185, SAMN02677186, SAMN02677187, SAMN02677195, SAMN00710542. Finally, the following publicly available genomes are available on NCBI and were used as references in this study: P. falciparum mitochondria: LR605957.1, P. vivax mitochondria: LT635627.1, P. malariae mitochondria: LT594637.1, P. falciparum nuclear chromosomes: GCA_000002765.3, P. vivax nuclear chromosomes: GCA_900093555.1, P. vivax-like nuclear chromosomes: GCA_003402215.1, Plasmodium cynomolgi nuclear chromosomes: GCA_900180395.1, P. praefalciparum nuclear chromosomes: GCA_900095595.1, and the Genome Reference Consortium Human Build 37 (HG19): PRJNA31257.

## Research involving human participants, their data, or biological material

Policy information about studies with human participants or human data. See also policy information about sex, gender (identity/presentation), and sexual orientation and race, ethnicity and racism.

| | |
|---|---|
| Reporting on sex and gender | All references to the sex of ancient individuals refers to biological sex, as determined based on osteological and/or genetic analysis. |
| Reporting on race, ethnicity, or other socially relevant groupings | NA |
| Population characteristics | NA |
| Recruitment | NA |
| Ethics oversight | NA |

Note that full information on the approval of the study protocol must also be provided in the manuscript.

# Field-specific reporting

Please select the one below that is the best fit for your research. If you are not sure, read the appropriate sections before making your selection.

☒ Life sciences          ☐ Behavioural & social sciences          ☐ Ecological, evolutionary & environmental sciences

For a reference copy of the document with all sections, see nature.com/documents/nr-reporting-summary-flat.pdf

# Life sciences study design

All studies must disclose on these points even when the disclosure is negative.

| | |
|---|---|
| Sample size | No sample size calculation was performed in association with this study. Instead, we used a metagenomic approach to identify candidate libraries preserving ancient Plasmodium DNA and captured all those exceeding preservational thresholds. |
| Data exclusions | No data was excluded from this study. |

| Replication | For population genetic analyses such as ADMIXTURE and phylogenetics, we used bootstrap replicates to infer confidence estimates for inferred relationships. |
| Randomization | For the Bayesian phylogenetic dating, we performed a date randomization test to compare the mutation rates estimated using randomly shuffled dates compared to true sampling dates for Plasmodium vivax strains. Out of 15 replicate BEAST analyses with shuffled dates, all 15 yielded mutation rate estimates with a 95% highest posterior density interval overlapping the 95% highest posterior density interval inferred using true dates, indicating that there is insufficient temporal signal in the dataset to reliably date divergence events. |
| Blinding | No blinding was performed in the present study. We performed only quantitative genomic analyses which are less vulnerable to subjective biases. Furthermore, as the samples analyzed derive from ancient archaeological remains that exhibited some evidence of malaria infection, they could not be allocated into experimental groups for which blinding would be an appropriate experimental design. |

# Reporting for specific materials, systems and methods

We require information from authors about some types of materials, experimental systems and methods used in many studies. Here, indicate whether each material, system or method listed is relevant to your study. If you are not sure if a list item applies to your research, read the appropriate section before selecting a response.

## Materials & experimental systems

| n/a | Involved in the study |
|---|---|
| ☒ | ☐ Antibodies |
| ☒ | ☐ Eukaryotic cell lines |
| ☐ | ☒ Palaeontology and archaeology |
| ☒ | ☐ Animals and other organisms |
| ☒ | ☐ Clinical data |
| ☒ | ☐ Dual use research of concern |
| ☒ | ☐ Plants |

## Methods

| n/a | Involved in the study |
|---|---|
| ☒ | ☐ ChIP-seq |
| ☒ | ☐ Flow cytometry |
| ☒ | ☐ MRI-based neuroimaging |

## Palaeontology and Archaeology

| Specimen provenance | Samples were obtained with prior permission from the appropriate authority in each case and following legal guidelines in the relevant country/countries. |
| Specimen deposition | Sequences/genotypes associated with this project have been deposited on the ENA and/or figshare to facilitate access by other researchers. |
| Dating methods | New dates were generated by AMC C14 dating at the Mannheim lab. All dates were recalibrated using the OxCal software (IntCal20 calibration curve). |

☒ Tick this box to confirm that the raw and calibrated dates are available in the paper or in Supplementary Information.

| Ethics oversight | Prehistoric and historic individuals, therefore no ethical oversight was required. |

Note that full information on the approval of the study protocol must also be provided in the manuscript.

## Plants

| Seed stocks | *Report on the source of all seed stocks or other plant material used. If applicable, state the seed stock centre and catalogue number. If plant specimens were collected from the field, describe the collection location, date and sampling procedures.* |
| Novel plant genotypes | *Describe the methods by which all novel plant genotypes were produced. This includes those generated by transgenic approaches, gene editing, chemical/radiation-based mutagenesis and hybridization. For transgenic lines, describe the transformation method, the number of independent lines analyzed and the generation upon which experiments were performed. For gene-edited lines, describe the editor used, the endogenous sequence targeted for editing, the targeting guide RNA sequence (if applicable) and how the editor was applied.* |
| Authentication | *Describe any authentication procedures for each seed stock used or novel genotype generated. Describe any experiments used to assess the effect of a mutation and, where applicable, how potential secondary effects (e.g. second site T-DNA insertions, mosiacism, off-target gene editing) were examined.* |

