## [Peer Review file · Nature]

Manuscript Title: Ancient Plasmodium genomes shed light on the history of human malaria

Reviewer Comments & Author Rebuttals

Reviewer Reports on the Initial Version:

Referees' comments:

Referee #1 (Remarks to the Author):

This manuscript from Michel and colleagues presents an intriguing and unprecedented report on the nuclear and mitochondrial genome sequences of *Plasmodium falciparum*, *Plasmodium malariae*, and *Plasmodium vivax* malaria parasites from ancient DNA samples, obtained from 36 infected skeletal remains spanning 16 countries and 5500 years. The median per-sample coverage was not impressive by standards applicable to contemporary samples, but this nevertheless represents a technical tour de force.

This work reports several important findings. Most importantly, it provides the most compelling evidence yet that *P. vivax* populations in the Americas most likely represent a now-extirpated European parasite population that crossed the Atlantic during the colonial era. This manuscript will be cited frequently for finally bringing robust evidence to support that assertion.

Other findings are novel given the unprecedented nature of the data, but not unexpected: the ancient occurrence of *Pv* and *Pf* infections in Europe, or the capacity of transitional hunter-gatherer populations to sustain *P. vivax* transmission, or the evidence of cross-cultural connectivity from the Chokhopani individual. The analyses of phylogeographic structure in contemporary parasite populations are not novel, nor the observation of differences in diversity between *Pf* and *Pv*, or the inference that *Pf* populations in the Americas were seeded by African populations.

The manuscript is clearly written, and the analyses appear to be appropriate and carefully conducted. The manuscript could potentially be improved through attention to a few small issues:

If American *P. vivax* populations largely represent an extirpated European parasite population, what can we infer from the magnitude of divergence of the American/European parasite clade from other parasite populations in terms of the age of that parasite population? Was *P. vivax* likely infecting neandertals in Europe, for example?

Figure 2: Could Latin America *P. vivax* be an admixed population representing an extirpated European parasite clade as well as a Western Asian population? This hypothesis is not given much attention in the text, but the ADMIXTURE analysis (panel c) suggests this possibility. Western Asian peoples were brought to the Americas as indentured servants during the colonial era, in large numbers in certain regions (eg Guyana, Trinidad & Tobago).

Minor notes:

the term 'eradication' is typically used to connote global extirpation of a disease; 'elimination' is favored for local extirpation.

ENA accession numbers are lacking

Referee #2 (Remarks to the Author):

I've read the Plasmodium manuscript by Michel et al with interest as it is well written, informative and reminded me of several talks I've attended at the Sanger Institute about Plasmodium. First scanning large existing databases for any Plasmodium signals (data mining) is furthermore a smart approach. These analyses show that *P. vivax* was already present in Eurasia for a very long time. Ancient DNA libraries were however not really needed to conclude that slave trade was the likely cause of *P. falciparum* in South America though they strengthen the point as none of the ancient strains look anything similar to modern SA strains. So, the main challenge is to make a case that *P. vivax*, as found in Latin America, is indeed from Europeans.

The tree shown in Figure 2B in combination with Extended Figure 9 leads me to believe there could have been 2 to 3 introductions of *P. vivax* in South America (unlike with *P. falciparum* which just looks like 1 plain introduction from Africa). Does this seem plausible?

Can you now, with these additional isolates, make a better prediction of the introduction of *P. vivax* into America (like they did here in Figure 3:

<https://academic.oup.com/mbe/article/37/3/773/5614438?login=false>)? A similar analysis (for comparison's sake) might be nice for *P. falciparum* as well. Does the introduction of *P. vivax* precede the introduction of *P. falciparum* in South America? Can it be said with certainty that *P. vivax* was absent from America before 1500? Absence of resistance alleles seems to be indicative (the only populations with sizeable double duffy negativity in Colombia e.g. are of African descent: <https://www.ncbi.nlm.nih.gov/pmc/articles/PMC4001950/>). Can the age of LDC020 be estimated even better btw? If it was before the Spanish contacts (around 1540(?)), then it would seem unlikely that Europeans brought it.

Extended data figure 1a: There is a STR isolate which seems more closely related to the Latin American Group than Ebr01944 yet I do not see it getting much attention. Is it perhaps a low coverage strain, and even if so, could you provide more information on it? Which STR isolate is it? STR105? In Extended data Figure 1b and 1c I furthermore see STR015 in 1b and STR105 in 1c, is 105 an error?

Minor points:

It would perhaps be a good idea to also show the *P. vivax* isolate names mentioned in Figure 2 also in Figure 1. This would allow me to at least interpret the connection between both figures a bit better.

The same is true in Extended data figure 1a. Show the names of isolates close to the South American group.

Similarly, in 4a it might be nice to put a few names to the modern background samples (where each of the clouds of gray dots is nowadays largely associated with).

Duffy double negative is near fixation in Africa but is also somewhat elevated in Germany as well: <https://www.ncbi.nlm.nih.gov/pmc/articles/PMC3074097/> . Is this perhaps something worth discussing in light of some of your more ancient findings?

Referee #3 (Remarks to the Author):

This manuscript describes genetic analyses of 36 ancient *Plasmodium* strains retrieved from skeletal material, along with ancestry analysis of some of the human individuals hosting those pathogens. To my knowledge, this is the most comprehensive study on the history of malaria parasites using ancient DNA techniques. Michel and colleagues screen shotgun data from around 10,000 ancient individuals and found 36 cases of malaria. After in-solution capture, they recover mitochondrial genomes at high coverage and nuclear data at low coverage, with most samples having a few thousand segregating SNPs covered for population analyses. I highlight this not as a critique, rather the contrary, to underscore the big technical challenges for recovering good-quality nuclear *Plasmodium* DNA from ancient samples. Although I am not an expert on *Plasmodium* genetics, overall the findings reported in the paper appear to be solid and are based on appropriate statistical analyses.

The authors reach three main conclusions:

- 1- *Plasmodium vivax* was present in Eurasia at least since the 4th millennium BC and in one case infecting a transitional hunter-gatherer population. This is, in my opinion, a novel result that significantly changes the current view about the geographical distribution of malaria parasites in the past.
- 2- A strong similarity between American and extinct European *P. vivax* strains, strongly suggesting that *P. vivax* was introduced in the Americas by European colonizers. This was already observed with the sequencing of the Ebro1944 strain (Gelabert et al. PNAS 2016 and Van Dorp et al. MBE 2020), but it is now further supported by additional European strains.
- 3- A strong similarity between American and African *P. falciparum* strains, strongly suggesting that *P. falciparum* was introduced into the Americas from Africa during the trans-Atlantic slave trade colonizers. This is based entirely from analysis of previously published modern strains, while the newly reported ancient strains from Europe support this result by excluding Europe as a plausible source of American *P. falciparum* strains.

An important section of the paper is devoted to the description of malaria cases at unexpected locations, which give insights into human mobility patterns driving malaria dissemination. My main comment is related to the human DNA analysis within this section. I am aware that the paper's main focus is the study of *Plasmodium* genetic patterns, but the human DNA data is insufficiently

analysed. For instance, the ancestry analysis of sample LDC020 is only dedicated to demonstrate a fully Native American profile lacking European admixture. However, with ~1 million 1240k SNPs covered, it should be possible to achieve much higher resolution. Is this individual's ancestry deriving from the Andean highlands or from Amazonian populations? The same goes for the human ancestry analysis of the Malaria cases at Mechelen, where the authors simply run a PCA on Western Eurasian populations and conclude that individuals during the earlier period "form a tight cluster overlapping modern French, English Scottish, and Hungarian population" while in the later periods "13 of 15 individuals are male and display a heterogeneous ancestry profile consistent with origins from across the Mediterranean". I am convinced that the excellent data quality for these individuals allows one to narrow down their ancestral origins at higher resolution than just the Mediterranean region:

-Do any of the Malaria cases have Iberian ancestry? Do the three individuals plotting in PCA close to the Sardinia Medieval samples most likely derive from Sardinian population? Or is their shift towards the bottom right part of the plot caused by African ancestry? To answer these questions, one needs to apply other techniques beyond PCA, such as ancestry modelling.

-Do the two women from the later periods have local ancestry? I cannot tell from the available information. An interesting observation is that all the *P. falciparum* and *P. malariae* cases have non-local ancestry, which is not explicitly mentioned in the text.

-Y-chromosome haplogroups tend to be highly structured and could provide additional clues for studying the place of origin of male individuals, but Y-chromosome information is lacking throughout the text and supplementary tables.

-Similarly, IBD analysis could reveal recent connections between Mechelen individuals and previously published ancient individuals from other regions.

-Finally, are there any families at Mechelen? If so, do you find several malaria cases within the same families?

I believe that a more detailed analysis of these individuals' ancestral origins could result in a better understanding of the routes of malaria spread, as well as its underlying causes. This should be fairly straightforward to implement as some of the authors are experts in the analysis of ancient human data.

Minor comments

-In Figure 4a, it would be useful to include published medieval individuals from Iberia, given that Spain was one of the main sources for soldiers in the Army of Flanders.

-In lines 417-419, the sentence is confusing because it means that all the 13 men from the middle and late phases are consistent with origins from across the Mediterranean, which is not true because according to the PCA, four individuals from these phases have local central European ancestry and at least two of them must be men.

-In Extended Data Figure 1a, to identify which symbol refers to the Egyptian strain, I need to go to Table S1 and get the site ID for the Egyptian site. Including the sample IDs, or maybe just the site IDs in the map of Figure 1 would be helpful for understanding the geographical patterns of the ancient samples in PCAs and other figures, given that they are coloured by chronology and not geography.

-In Figure 4c, the X axis is labelled x.rate and the Y axis is not labelled.

-In the legend of Extended Data Figure 5b-c, please indicate whether the Native American reference groups have any European admixture.

Referee #4 (Remarks to the Author):

This manuscript presents novel data from an array of ancient mitochondrial and nuclear genomes of three Plasmodium species that produce malaria in humans. Results show that Africa was not the source of both *P. falciparum* and *P. vivax* in the Americas after European contact and forced importation of African slaves. Instead, European colonizers were themselves the source of *P. vivax* infections. This work adds substantially to our understanding of Plasmodium evolution and malaria in past human populations.

It was a pleasure to read such a thorough and well-written manuscript. The sample size is excellent for a study of this kind and could be increased strategically with future work to resolve sampling gaps and investigate specific questions. My expertise is not in genetics, so I cannot comment on the methods used. I have only a few questions and suggestions for improvement.

p. 4-5, lines 139-148: In the discussion of Duffy antigen negativity, there is no mention made of the different mutation found in Papua New Guinea. Doesn't it offer some protection against malaria as well? How does its presence affect the African origin hypothesis?

On p. 9, line 241-243, three sites with *P. falciparum* genomic data are mentioned with inclusion of calibrated radiocarbon dates, yet there are no citations for this specific information (which need to be included in the main text and not just supplemental material). Additionally, no reference to the supplemental information is provided to direct the reader to the archaeological context. Similarly, sites with specific dates are indicated in lines 254-256 and lack citations and reference to the supplemental material on the archaeological context. More information and citations on Gundorovka, particularly, should be incorporated in the text to substantiate the claim that it is a low-density, transitional hunter-gather population. Readers should not have to wade through supplemental material to find such supporting information.

At the bottom of p. 14, Chokhopani is discussed again. It is desirable to include very briefly in the text some basic information about this site, such as the size of the settlement area and cemetery rather than having it only in the supplemental information. Please also reference Supplementary

Note 1.3 here for the specific information about this site and area. Additionally, on p. 15, line 376, sex and age estimations at a minimum should be included in the text for the individual with *P. vivax*. One should not have to dig through the supplemental information to learn the person is genetically male and still without any indication of age. Without even basic information about this individual, the statement at the end of the paragraph that their disease status adds to this person's osteobiography is overblown. The possibility that this person was an immigrant from a nearby endemic area rather than a resident who traveled to such an area and returned should also be considered on p. 15, lines 381-382.

At St. Rombout, in Mechelen, Belgium, on p. 16, line 412 on, what does co-infection with more than one species of *Plasmodium* suggest in terms of impact on the infected individuals, mobility, etc.? A little more explanation is needed. At the end of the paragraph on p. 17 lines 424-426, the last statement suggests that malaria was transmitted by infected soldiers to the local population. Were there any mosquito species present to transmit *P. falciparum* in the Low Countries during this time? This critical vector information is provided for Chokhopani and Chachapoyas areas but is lacking for Mechelen. Do you mean to say here that the infected soldiers contracted it elsewhere and died without transmitting it via *Anopheles* mosquitoes to the local populace? Please clarify. This issue arises again in the Conclusions and Implications section, lines 434-436. Invoking "human mobility in spreading malaria" suggests it was transmitted to local people, rather than simply being carried by the infected person to the area and dying out due to absence of mosquito vectors for *P. falciparum* in the Himalayan highlands and temperate Europe. The latter conclusion is supported by the evidence provided but the authors do not demonstrate that malaria spread to others. I suggest rephrasing here to "...human mobility in carrying malaria within them to peripheries of endemic zones." If I have misunderstood, please clarify.

As a minor editorial note, recall that data are plural (datum is singular). There is disagreement on line 109 ("data underscore" not "underscores"), line 277 and 431 (both should be "data provide"), and line 698 (should read "data were"). I noticed a couple of instances of such disagreement in the supplemental information file as well but did not record them.

On line 113, "case-study" should be two separate words without a hyphen.

Author Rebuttals to Initial Comments:

It is our pleasure to submit the requested revisions to our manuscript, “**Ancient *Plasmodium* genomes shed light on the history of human malaria**” (#2023-07-12551A).

We thank the reviewers for their comments and suggestions, which we have addressed in detail below (see Point-By-Point Response). The resulting revised manuscript presents substantial new analyses, which add to our understanding of malaria transmission in the past at both global and regional scales.

Following the comments of Reviewer 3, we performed additional analyses to shed light on the human population genetics of individuals from the sites of Laguna de los Cóncores, Peru and Mechelen, Belgium. Using ancestry modeling, we demonstrate affinity between the malaria-infected individual from Laguna de los Cóncores and Indigenous populations from the Peruvian coast and highlands. For Mechelen, we modeled ancestry and inferred geographic origin using spatiotemporal interpolation of human genetic ancestry for each of the ten malaria-positive individuals. The new analyses support and refine our original observation that the individuals from the later phase of the cemetery co-infected with different malaria strains display increased genetic affinity to the Mediterranean, especially with southern Spain and Italy and, to a lesser extent, with the Near East. Together, these findings strengthen our claims regarding the role of human mobility in spreading malaria during the colonial and Early Modern periods.

Prompted by questions from Reviewers 1 and 2, we undertook additional analyses to investigate whether modern Latin American *P. vivax* populations likely descend from a single European source or derive from multiple introductions and/or admixture events. Interestingly, we identify clear substructure in Latin American *P. vivax* populations, which provides evidence for a complex population history beyond a single source introduction. Furthermore, we present new analyses showing that the pericontact Peruvian strain LDC020 is more closely related to modern strains from Peru than other modern Latin American subpopulations, suggesting that traces of this pericontact ancestry persist in the region today. This finding supports establishment of an endemic malaria focus in the remote Peruvian highlands soon after European contact, illuminating the processes by which new pathogens gained a foothold in the Americas during the period of the Columbian Exchange.

Finally, in response to questions from Reviewers 1 and 2, we perform Bayesian molecular dating using BEAST to explore the divergence between Latin American and European *P. vivax* populations. Due to low data coverage and challenges posed by recombination, we find significant uncertainty in population split times. Nevertheless, we present the results of these analyses alongside a detailed discussion of the challenges associated with molecular dating analyses in *P. vivax*.

Together, we believe that the new results and discussion have significantly strengthened the manuscript, and we are grateful to the editor for the opportunity to submit this revised study.

Yours sincerely,
Megan Michel, Johannes Krause, and Alexander Herbig

Point-by-Point Response to Reviewer Comments for MANUSCRIPT NUMBER 2023-07-12551A
“Ancient *Plasmodium* genomes shed light on the history of human malaria”

We received 38 specific comments from four reviewers. We present our responses to each reviewer comment below.

Reviewer #1 (Remarks to the Author):

Comment 1: *If American *P. vivax* populations largely represent an extirpated European parasite population, what can we infer from the magnitude of divergence of the American/European parasite clade from other parasite populations in terms of the age of that parasite population? Was *P. vivax* likely infecting neandertals in Europe, for example?*

In order to explore the age of European and Latin American *P. vivax* lineages, we performed molecular dating on two different datasets using the software BEAST (Bayesian Evolutionary Analysis by Sampling Trees). While both datasets include the high coverage ancient *P. vivax* samples Ebro1944 and LDC020, they differ in their complement of modern comparative strains. Unfortunately, we encountered significant methodological difficulties in both analyses, primarily related to (1.) the low coverage and relatively recent date of the ancient samples, and (2.) the lack of satisfactory methods for dealing with recombinant positions in *Plasmodium* nuclear SNP alignments. As a result, we find that we are unable to infer a reliable date for the divergence of the American and European parasite clades. Nevertheless, we summarize our methods and the results of our BEAST analyses, highlighting both the challenges that we encountered and outlining potential areas for future development (see **Supplementary Methods 12: Bayesian Phylogenetic Analysis and Molecular Dating** and **Supplementary Note 13: Bayesian Molecular Dating Using Beast**, as well as **Extended Data Figure 11** and **Supplementary Table 16: Date Randomization Test**).

Comment 2: *[Referring to data presented in] Figure 2, could Latin America *P. vivax* be an admixed population representing an extirpated European parasite clade as well as a Western Asian population? This hypothesis is not given much attention in the text, but the ADMIXTURE analysis (panel c) suggests this possibility. Western Asian peoples were brought to the Americas as indentured servants during the colonial era, in large numbers in certain regions (eg Guyana, Trinidad & Tobago).*

We thank the reviewer for their question regarding potential admixture events in Latin American *P. vivax* populations and present additional population genetic analyses to clarify this point. First, we used principal component analysis to identify phylogeographic substructure within Latin American *P. vivax* (**Extended Data Figure 1b**). To test for affinity between Latin American *P. vivax* subpopulations and modern Eurasian populations (including Western Asian lineages), we utilize F_4 statistics of the form $f_4(\text{LAM1, LAM2; WAS/AF/ETH, } P. \text{ vivax-like})$ (**Supplementary Table 5: LAM *P. vivax* F_4 Statistics**). These results confirm that Latin American *P. vivax* strains do not represent a homogenous population, and that strains from Colombia, Peru, and Brazil exhibit an excess of ancestry related to WAS, AF, and ETH. This finding would be consistent with admixture between a subset of Latin American populations and a source related to WAS/AF/ETH. We present and contextualize these results in a new supplementary note (**Supplementary Note 6: *P. vivax* Population Structure within Latin America** and **Supplementary Table 5: LAM *P. vivax* F_4 Statistics**, see also updates in the main manuscript **Methods: Plasmodium Population Genetic Analysis**, paragraph 5).

Comment 3: *The term ‘eradication’ is typically used to connote global extirpation of a disease; ‘elimination’ is favored for local extirpation.*

Throughout the text, we have replaced the term ‘eradication’ with ‘elimination’ where we refer to local extirpation of malaria.

Comment 4: *ENA accession numbers are lacking.*

ENA numbers have been added, and data will be released upon publication of the manuscript.

Reviewer #2 (Remarks to the Author):

Comment 1: *The tree shown in Figure 2B in combination with Extended Figure 9 leads me to believe there could have been 2 to 3 introductions of *P. vivax* in South America (unlike with *P. falciparum* which just looks like 1 plain introduction from Africa). Does this seem plausible?*

While the hypothesis of multiple introductions of *P. vivax* into Latin America is highly plausible, we argue that such events cannot be reliably inferred from the phylogeny presented in Figure 2b, which exhibits low bootstrap values and uncertainty in tree topology. To further explore the hypothesis of multiple introductions, we computed F_{ST} -statistics of the form $f_4(\text{LAM1}, \text{LAM2}; \text{TEST}, P. \text{vivax-like})$ for all pairs of modern Latin American subpopulations (**Supplementary Table 5: LAM *P. vivax* F4 Statistics**). If Latin American strains derive from a single introduction event, we expect all pairs of LAM strains to be symmetrically related to a global set of modern *P. vivax* populations as well as the Ebro1944 European strain. In fact, we find evidence for distinct ancestry profiles in Latin American *P. vivax* subpopulations, consistent with multiple introduction events and/or subsequent admixture events in a subset of LAM subpopulations. We discuss these results in detail in a new supplementary note (**Supplementary Note 6: *P. vivax* Population Structure within Latin America**, see also updates in the main manuscript **Methods: Plasmodium Population Genetic Analysis**, paragraph 5). While a similar set of analyses could be used to compare this history to that of South American *P. falciparum* populations, our current dataset is limited to only 18 *P. falciparum* strains from two South American countries (Peru and Colombia). We argue that this limited sampling precludes analyses of South American *P. falciparum* population substructure, and further inferences must await additional sampling of modern strains from the region.

Comment 2: *Can you now, with these additional isolates, make a better prediction of the introduction of *P. vivax* into America (like they did here in Figure 3: [van Dorp et al. 2020])?*

To explore whether inclusion of the new high-coverage ancient strain LDC020 would enable more accurate dating of the LAM *P. vivax* clade, we attempted two molecular dating analyses using the software BEAST. As discussed, we encountered significant methodological difficulties in both analyses, primarily related to 1.) the low coverage and relatively recent date of our ancient samples, and 2.) the lack of methods for dealing with recombinant positions in *Plasmodium* nuclear SNP alignments. Regarding the latter point, we attempted to replicate the procedure employed by van Dorp et al. 2020 to remove recombinant SNPs. However, we found that such an approach to homoplasy removal has a differential impact on clades across the *P. vivax* phylogeny, potentially leading to biases in the inferred clock rate. Although we attempted to circumvent this issue by selecting single strains from geographically isolated (and therefore not currently recombining) global populations, we observe inconsistencies between the inferred root dates and the population genetics of our low-coverage ancient strains. Therefore, we conclude that our dataset is not suitable for reliably inferring the age of the Latin American *P. vivax* clade. Nevertheless, we summarize these results in a new supplementary note, highlighting the challenges that we encountered (see **Supplementary Methods 12: Bayesian Phylogenetic Analysis and Molecular Dating** and **Supplementary Note 13: Bayesian Molecular**

Dating Using Beast, as well as Extended Data Figure 11 and Supplementary Table 16: Date Randomization Test).

Comment 3: *A similar analysis (for comparison's sake) might be nice for P. falciparum as well. Does the introduction of P. vivax precede the introduction of P. falciparum in South America?*

As discussed above, we encountered significant difficulties in obtaining accurate divergence date estimates for nodes in the *P. vivax* phylogeny. Furthermore, we are able to capture a much smaller proportion of the *P. falciparum* nuclear genome (53.6%, compared to 99.8% of the *P. vivax* nuclear genome), and we obtained lower genomic coverage even for our best-preserved *P. falciparum* strains. As a result, the current dataset lacks high-coverage ancient samples suitable for calibrating the *P. falciparum* phylogeny. Combined with the reduced intraspecific diversity present in modern *P. falciparum* populations, we argue that our low-coverage dataset is insufficient for performing the requested divergence dating in this species.

Comment 4: *Can it be said with certainty that P. vivax was absent from America before 1500? Absence of resistance alleles seems to be indicative (the only populations with sizeable double duffy negativity in Colombia e.g. are of African descent: <https://www.ncbi.nlm.nih.gov/pmc/articles/PMC4001950/>).*

We thank the reviewer for raising this important point. While we do not think that our results definitively answer this question, we argue that our findings are consistent with the absence of *Plasmodium* spp. in the Americas prior to the contact period. We have added several sentences in the **Introduction** (page 7) and section on **Alternative Histories in the Americas** (page 14) to better address these points. See text below:

page 7- “A contact-era introduction of *Plasmodium* spp. is consistent with the absence of malaria resistance alleles in Indigenous peoples of the Americas (Hedrick 2011).”

page 14- “Overall, this evidence for a close genetic link between American and extirpated European strains suggests that *P. vivax* was likely absent in the Americas prior to the contact period, although we cannot exclude the possibility of a replacement of pre-contact *P. vivax* variation following the introduction of strains from Europe.”

Comment 5: *Can the age of LDC020 be estimated even better btw? If it was before the Spanish contacts (around 1540(?), then it would seem unlikely that Europeans brought it.*

Unfortunately, although we generated C14 dates from two separate teeth from individual LDC020, the 95% confidence intervals for the calibrated dates are non-overlapping. Moreover, one calibrated date spans the contact period while the other does not. Similar cases of incompatible C14 dates have been previously observed in pairs of related individuals, possibly due to contamination of C14 dated samples (for example, see Sedig *et al.* 2021). While the cause of the anomaly observed in this case is unclear, challenges associated with radiocarbon dating in the pericontact period are mentioned elsewhere (see Wild *et al.* 2007). Similarly, as outlined in published materials, the site's use period spans the contact period (Wild *et al.* 2007), and disruption of the archaeological context through extensive looting prevents differentiation of pre- and post-contact contexts. To better reflect this uncertainty, we have expanded our discussion of the archaeological context to highlight these difficulties in radiocarbon and contextual dating. The following sentences in **Supplementary Note 1: Archaeological Background, 2.6 Laguna de los Cóndores** have been added/updated:

2.6 Laguna de los Cóndores- Discovered by ranch hands in 1996, extensive looting impacted 90% of the site before rescue excavations began the following year (von Hagen and Guillén 1998) ... Assessment of material culture and radiocarbon dating of human remains and

organic artifacts indicate that the *chullpas*' use period began prior to the Incan conquest (Late Intermediate Period, 1000-1475 CE), continuing through the Incan occupation (Late Horizon, 1475-1532 CE) and into the Spanish colonial era (von Hagen and Guillén 1998, Wild *et al.* 2007). Indeed, recovery of colonial-era artifacts including a Christian crucifix supports the site's continued use post-contact. However, the short duration of the Late Horizon and unfavorable form of the calibration curve complicate attempts to establish chronology through radiocarbon dating (Wild *et al.* 2007); similarly, establishment of a contextual archaeological chronology has been challenging due to the extensive looting that took place prior to the onset of archaeological excavations.

- **LDC020 (CHA99)**- We recovered genome-wide *P. vivax* mitochondrial and nuclear data from two teeth from one genetically male individual from Laguna de los Cóndores. The skeletonized remains of this individual were commingled with others as a result of looting.

Comment 6: *Extended data figure 1a: There is a STR isolate which seems more closely related to the Latin American Group than Ebro1944 yet I do not see it getting much attention. Is it perhaps a low coverage strain, and even if so, could you provide more information on it? Which STR isolate is it? STR105?*

The STR isolate to which the reviewer is referring (STR067) is indeed a low coverage strain, with only 2,303 segregating target SNPs. As such, we suggest that the precise positioning of this isolate in PCA space should not be overinterpreted. To make this point apparent to the reader, we have added a bold outline to symbols representing high coverage ancient strains in **Extended Data Figure 1a**.

Comment 7: *In Extended data Figure 1b and 1c I furthermore see STR015 in 1b and STR105 in 1c, is 105 an error?*

We have corrected STR015 in **Extended Data Figure 1b** to the correct ID (STR105).

Comment 8: *It would perhaps be a good idea to also show the *P. vivax* isolate names mentioned in Figure 2 also in Figure 1. This would allow me to at least interpret the connection between both figures a bit better.*

We agree with the reviewer's suggestion and have included the names and abbreviations for sites yielding high coverage data in **Figure 1**.

Comment 9: *The same is true in Extended data figure 1a. Show the names of isolates close to the South American group.*

We have added site names in the legend of **Extended Data Figure 1a** to make distinguishing specific strains easier.

Comment 10: *Similarly, in 4a it might be nice to put a few names to the modern background samples (where each of the clouds of gray dots is nowadays largely associated with).*

In **Figure 4a**, we have added labels to the modern populations that are explicitly mentioned in the text and/or discussed in our updated population genetic analysis (for details, see **Supplementary Note 10: Human Population Genomic Analysis of Individuals from Mechelen, Belgium**).

Comment 11: *Duffy double negative is near fixation in Africa but is also somewhat elevated in Germany as well: <https://www.ncbi.nlm.nih.gov/pmc/articles/PMC3074097/>. Is this perhaps something worth discussing in light of some of your more ancient findings?*

In light of the evidence from ancient *Plasmodium* genomes, we agree with the reviewer's suggestion that investigating the temporal history and distribution of malaria resistance alleles in European populations represents an exciting avenue for future study. However, upon review of both the cited paper (Howes *et al.* 2011) and subsequent literature on the global distribution of the Duffy blood group alleles, we argue that there is not sufficient evidence for elevated frequencies of the Duffy null variant in Germany to merit further discussion in the manuscript. Frequency estimates for the FY*B^{ES} allele presented by Howes *et al.* are based on a meta-analysis of previous studies, some of which suffered from small sample sizes and most of which utilized phenotypic rather than genotypic data. As a result, the apparent elevated frequency of the FY*B^{ES} variant in Germany is associated with a high degree of uncertainty, as reflected in the large interquartile range of the allele frequency posterior distributions for those regions (see Howes *et al.* 2011, Fig. 3c and Fig. 3f). Even assuming accuracy in genotype estimates, the predicted frequency of FY*B^{ES} homozygotes exhibiting the Duffy-negative phenotype is quite low in populations outside of Africa (e.g. reaching a maximum of 10-20% for parts of Germany). Finally, Howes *et al.* do not seem to have accounted for the ancestry of study participants whose Duffy genotype/phenotype data was measured, making it possible that study participants represented recent immigrants to Europe. A subsequent study of whole-genome sequencing data from the HGDP and 1000 Genomes Project including over 500 individuals of European ancestry found negligible frequencies of the FY*B^{ES} allele in European and Asian populations (McManus *et al.* 2017).

Reviewer #3 (Remarks to the Author):

Comment 1: *My main comment is related to the human DNA analysis within this section. I am aware that the paper's main focus is the study of Plasmodium genetic patterns, but the human DNA data is insufficiently analysed. For instance, the ancestry analysis of sample LDC020 is only dedicated to demonstrate a fully Native American profile lacking European admixture. However, with ~1 million 1240k SNPs covered, it should be possible to achieve much higher resolution. Is this individual's ancestry deriving from the Andean highlands or from Amazonian populations?*

We thank the reviewer for their suggestion to investigate this aspect in more depth. This question is particularly interesting and relevant in light of our new evidence for affinity between LDC020 and modern Peruvian *P. vivax* populations, and we have undertaken additional population genetic analyses for individual LDC020. In the revised manuscript, we show using PCA and *qpWave* modeling that LDC020 is consistent with deriving his ancestry from ancient highland or coastal populations of present-day Peru. This evidence for genetic continuity in the region parallels our findings from pathogen genomics. We present these new analyses and contextualize the results in an updated supplementary note (**Supplementary Note 8: Human Population Genomics of LDC020**, see also **Supplementary Table 14: LDC020 Ancestry Modeling, Extended Data Figure 5**, and updates in **Supplementary Methods 10: Human Population Genetic Analysis**).

Comment 2: *The same goes for the human ancestry analysis of the Malaria cases at Mechelen, where the authors simply run a PCA on Western Eurasian populations and conclude that individuals during the earlier period “form a tight cluster overlapping modern French, English Scottish, and Hungarian population” while in the later periods “13 of 15 individuals are male and display a heterogeneous ancestry profile consistent with origins from across the Mediterranean”. I am convinced that the excellent data quality for these individuals allows one to narrow down their ancestral origins at*

higher resolution than just the Mediterranean region. Do any of the Malaria cases have Iberian ancestry?

Similar to LCD020, in our revised version we are presenting extensive analyses on the genome-wide data from the site of Mechelen (STR). First, we use the filtered output of PCA with spatiotemporal meta-information as input to a new method called *mobest*. This method computes a spatiotemporal interpolation of human genetic ancestry components and -based on the age of the test individual- it infers the similarity in probability space. The advantage of this method is that so-called ‘genetic outliers’ are taken into account in the interpolation, whereas in methods like *qpAdm* the user selects groups of individuals that best represent the ancestry in an area at a given time. On the other hand, *mobest* does not consider recent admixture events that could result in intermediate PC1-PC2 space. Therefore, we also applied *qpWave/qpAdm*, first asking which published groups from the last millennium with similar PC coordinates and high similarity probability in *mobest* can be modeled as sister clades to the STR individuals, and then advancing with more complex two-way models. Our results combined (**Supplementary Table 8: STR Ancestry Modeling and Extended Data Figure 6**) do suggest that 4 individuals (STR016, STR067, STR105, STR140) have predominantly Iberian ancestry, more specifically from Mediterranean present-day Spain (modeled either with medieval or present-day groups). We present these new analyses and contextualize the results in a new supplementary note (**Supplementary Note 10: Human Population Genomic Analysis of Individuals from Mechelen, Belgium**, see also updated methods in **Supplementary Methods 10: Human Population Genetic Analysis**).

Comment 3: *Do the three individuals plotting in PCA close to the Sardinia Medieval samples most likely derive from Sardinian population?*

Following the approach we present in our reply to comment 2, we could not conclude that any of these three individuals could have directly derived from Medieval Sardinia. We note here the sampling bias in the published data, since only four individuals are available from Medieval Sardinia, all of which are clearly distinct from present-day and Iron Age Sardinia, and exhibit high variation among them. However, for individual STR025 we show that a two-way admixture model between Late Medieval Germany and Medieval Sardinia provides a more adequate explanation of the data compared to the simpler model of 100% Medieval Spain (e.g., STR016, STR067, STR140). For details see **Supplementary Note 10: Human Population Genomic Analysis of Individuals from Mechelen, Belgium**, as well as **Supplementary Table 8: STR Ancestry Modeling, Extended Data Figure 6**, and updates in **Supplementary Methods 10: Human Population Genetic Analysis**.

Comment 4: *Or is their shift towards the bottom right part of the plot caused by African ancestry? To answer these questions, one needs to apply other techniques beyond PCA, such as ancestry modeling.*

We further explored this point by testing the performance of *qpAdm* models using data from the indigenous people of Guanches (‘CanaryIslands_Guanche.SG’) as a proxy of N. African ancestry. We show that models for STR067 and STR140 with a medieval group from Spain during the rule of the Carolingian dynasty (‘Spain_Carolingian’) reach the p-value threshold of ≥ 0.05 only when $\sim 10\%$ ancestry from CanaryIslands_Guanche.SG is added. However, other medieval groups from Spain (i.e., ‘Spain_Medieval’) can work as one-way models, supporting the hypothesis that low N. African-like ancestry was already assimilated in some medieval populations from Spain outside of the Islamic

contexts (Olalde *et al.*, 2019; citation is in the main text), and therefore STR067 and STR140 could have descended from them. For details see **Supplementary Note 10: Human Population Genomic Analysis of Individuals from Mechelen, Belgium**, as well as **Supplementary Table 8: STR Ancestry Modeling** and updates in **Supplementary Methods 10: Human Population Genetic Analysis**.

Comment 5: *Do the two women from the later periods have local ancestry? I cannot tell from the available information. An interesting observation is that all the *P. falciparum* and *P. malariae* cases have non-local ancestry, which is not explicitly mentioned in the text.*

The two genetically female individuals from the mid and late transects do indeed exhibit a local ancestry signal. We note that with the added **Extended Data Figure 6**, that provides individual ID labels on the PCA for the ‘non-locals’ and those infected with malaria, it is fairly easy to track such information now. We have also expanded our discussion in the main text as follows to include this observation, as well as the finding that all cases of *P. falciparum* and *P. malariae* infections derive from individuals with a non-local ancestry:

*page 18-19: “Compared to the early transect, 15 individuals recovered from the cemetery’s middle and late phases exhibit greater variation in both genetic ancestry and *Plasmodium* species detected. Of the 13 male individuals, 11 harbor heterogeneous ancestry encountered across the Mediterranean, while the remaining two females overlap the early phase cluster in PCA space...Remarkably, all individuals infected with *P. falciparum* and/or *P. malariae*, including the other two multispecies *Plasmodium* infections, exhibit non-local ancestry. ”*

Comment 6: *Y-chromosome haplogroups tend to be highly structured and could provide additional clues for studying the place of origin of male individuals, but Y-chromosome information is lacking throughout the text and supplementary tables.*

We apologize that this basic information was missing from our manuscript. Remarkably, 5/9 malaria-infected males -including ‘eastern’ STR091- belonged to the R-PF6547 branch (R1b1a1b1a1a2), which has been frequently encountered in Northwest and West Europe since the Iron Age. Interestingly, male STR025 belongs to a clade of H1a1a which is of S. Asian origin, but also commonly found today in Europe among Roma people, who had reached Europe by the time of Mechelen. This new information is presented/discussed in the following sections of the manuscript: **Supplementary Note 10: Human Population Genomic Analysis of Individuals from Mechelen, Belgium** and **Supplementary Methods 10: Human Population Genetic Analysis**.

Comment 7: *Similarly, IBD analysis could reveal recent connections between Mechelen individuals and previously published ancient individuals from other regions.*

We thank the reviewer for this suggestion to further refine our analyses of the human data. We explored our options and reckoned that a comprehensive IBD analysis would require uniform re-processing of all STR and other relevant published data from bam files all through genotype imputations and finally IBD estimations with ancIBD. We highlight that ibd segments shorter than 12cM can be maintained over hundreds of years and thus would be the most informative for our question, given that genomic data from 12th-17th c. AD across Europe are currently scarce. However,

these short segments are less reliable when computed from 1240K data. Therefore, we concluded that such an extensive analysis would be beyond the scope of this study, taking into account the resolution we have reached with the current analyses on the one hand, and the possible limitations with ibd on the other hand.

Comment 8: *Finally, are there any families at Mechelen? If so, do you find several malaria cases within the same families?*

We have now included analyses of biological relatedness estimated with four different methods (**Supplementary Methods 10: Human Population Genetic Analysis**). We found no close or more distant relatedness between any pairs of the presented individuals from Mechelen. For more information, see **Supplementary Table 15: Relatedness** and **Supplementary Note 10: Human Population Genomic Analysis of Individuals from Mechelen, Belgium**.

Comment 9: *In Figure 4a, it would be useful to include published medieval individuals from Iberia, given that Spain was one of the main sources for soldiers in the Army of Flanders.*

After including the new ancestry modeling and probability analyses, we have updated the annotation on **Figure 4a** providing the PC1 and PC2 coordinates of all the groups that are discussed throughout our analyses. We also plot several comparative ancient populations in Iberia in revised PCA (e.g. Spain Medieval, Spain Islamic, and Spain Carolingian).

Comment 10: *In lines 417-419, the sentence is confusing because it means that all the 13 men from the middle and late phases are consistent with origins from across the Mediterranean, which is not true because according to the PCA, four individuals from these phases have local central European ancestry and at least two of them must be men.*

We recognize that the phrasing in the main text is incorrect and could be misleading. The local population may have still used the cemetery during the middle and late phases, as suggested by the 4 individuals from the 15th-16th century transect (1400-1600) showing central European ancestry. In order to avoid confusion, we have updated the main text as follows:

*page 18- “Compared to the early transect, 15 individuals recovered from the cemetery’s middle and late phases exhibit greater variation in both genetic ancestry and *Plasmodium* species detected. Of the 13 male individuals, 11 harbor heterogeneous ancestry encountered across the Mediterranean, while the remaining two females overlap the early phase cluster in PCA space.”*

Comment 11: *In Extended Data Figure 1a, to identify which symbol refers to the Egyptian strain, I need to go to Table S1 and get the site ID for the Egyptian site. Including the sample IDs, or maybe just the site IDs in the map of Figure 1 would be helpful for understanding the geographical patterns of the ancient samples in PCAs and other figures, given that they are coloured by chronology and not geography.*

To enhance readability, we have added site names/abbreviations for samples explicitly mentioned in the main text in **Figure 1**. Furthermore, we include site names in the legend of **Extended**

Data Figure 1a to aid in identifying low-coverage samples (including the sample from Thebes) that are not explicitly mentioned in the main text.

Comment 12: *In Figure 4c, the X axis is labelled x.rate and the Y axis is not labelled.*

We have added clearer labels to both the X and Y axes in **Figure 4c**.

Comment 13: *In the legend of Extended Data Figure 5b-c, please indicate whether the Native American reference groups have any European admixture.*

We have updated the legend of **Extended Data Figure 5b-c** as follows to indicate that the Native American reference groups lack appreciable European admixture:

Extended Data Figure 5 legend- “a. PCA computed using a global set of modern human populations. LDC020 was projected onto these axes of variation (white star). b. F4-statistics testing for cladality between LDC020 and a test panel of modern South American Indigenous populations with negligible European admixture. A negative statistic indicates excess allele sharing between either Spanish.DG and the test population or LDC020 and Mbuti.DG, while a positive statistic supports excess affinity between either Spanish.DG and LDC020 or Mbuti.DG and the test population. Error bars show ± 3 standard errors. c. Supervised ADMIXTURE analysis modeling the ancestry of LDC020 as a composite of six modern populations: Atayal, French, Kalash, Karatiana, Mbuti, and Papuan. d. Regional PCA computed using select ancient and modern populations from the Americas (Supplementary Methods 10). LDC020 is projected onto these axes of variation.”

Reviewer #4 (Remarks to the Author):

Comment 1: *In the discussion of Duffy antigen negativity (p. 4-5, lines 139-148), there is no mention made of the different mutation found in Papua New Guinea. Doesn't it offer some protection against malaria as well? How does its presence affect the African origin hypothesis?*

The FY^*A^{null} allele found in Papua New Guinea occurs on a separate haplotype background compared to the FY^*B^{null} allele, which is nearly fixed in sub-Saharan African populations. While the FY^*A^{null} variant has been associated with resistance to erythrocyte invasion by *P. vivax* merozoites, it is thought that selection on this allele occurred much more recently (see Zimmerman *et al.* 1999). Not only is the allele present at much lower frequencies compared to the FY^*B^{null} variant, but it occurs on a long haplotype background suggestive of recent positive selection. Therefore, while the presence of Duffy negativity in Papua New Guinea is consistent with selection against *P. vivax* infection, we find that it does diminish the plausibility of the African origin hypothesis. We have modified the main text as follows to reflect this nuance:

page 5- “Together with the near-fixation of the Duffy negative allele in many human groups in sub-Saharan Africa, this provides strong support for an African origin for *P. vivax* (Carter and Mendis 2002). The Duffy antigen encoded by the *FY* locus facilitates *P. vivax* erythrocyte invasion, and individuals homozygous for the Duffy-negative allele were once considered completely immune to *P. vivax* malaria (Carter and Mendis 2002; Kwiatkowski 2005). While accumulating evidence demonstrates that populations with high rates of Duffy-negativity can in fact maintain low levels of *P. vivax* transmission, the phenotype appears to reduce erythrocyte invasion efficiency and to provide protection against blood-stage infection

(Twohig *et al.* 2019). Thus, proponents of the African origin hypothesis argue that a long history of selection pressure exerted by *P. vivax* drove increases in the Duffy-negative phenotype, making these populations less susceptible to *P. vivax* infection today. Interestingly, some human groups in Papua New Guinea harbor a Duffy null allele which appears to have arisen via an independent mutation; however, the very low frequency and long haplotype associated with the Papua New Guinea variant have been cited as evidence of more recent positive selection in Oceanians compared to African populations (Zimmerman et al. 1999).”

Comment 2: *On p. 9, line 241-243, three sites with P. falciparum genomic data are mentioned with inclusion of calibrated radiocarbon dates, yet there are no citations for this specific information (which need to be included in the main text and not just supplemental material). Additionally, no reference to the supplemental information is provided to direct the reader to the archaeological context. Similarly, sites with specific dates are indicated in lines 254-256 and lack citations and reference to the supplemental material on the archaeological context. More information and citations on Gundorovka, particularly, should be incorporated in the text to substantiate the claim that it is a low-density, transitional hunter-gather population. Readers should not have to wade through supplemental material to find such supporting information.*

To facilitate easier navigation of the supplementary materials, we have added pointers to the archaeological summaries for sites mentioned in the main text. We have also added citations in cases where radiocarbon analyses and/or contextual archaeological dating was published previously. In several other cases (e.g. Göttlesbrunn and Leubingen), we report new, unpublished radiocarbon dates in this study. To make that clearer, we have added a statement in the introduction to **Supplementary Note 1: Archaeological Background** indicating that radiocarbon dates not accompanied by a citation were generated as part of the present study. Finally, we added the following sentences and citation to provide additional context about the site of Gundorovka and the recovery of *P. vivax* from a population dated to the Eneolithic period:

page 10- “Evidence for *P. vivax* infection in the individual from Gundorovka is especially noteworthy; while the site’s use period spans the Neolithic-Eneolithic through the Middle-Late Bronze and Early Iron Ages, the individual analyzed here has been contextually dated to the Eneolithic period (Овчинникова and Хохлов 1998). Our findings underscore the need for additional sampling to fully elucidate the capacity of low-density transitional hunter-gatherer groups to sustain malaria transmission prior to the full-scale adoption of agriculture and sedentism.”

Comment 3: *At the bottom of p. 14, Chokhopani is discussed again. It is desirable to include very briefly in the text some basic information about this site, such as the size of the settlement area and cemetery rather than having it only in the supplemental information. Please also reference Supplementary Note 1.3 here for the specific information about this site and area.*

We thank the reviewer for raising this important point. We have added a reference to Supplementary Note 1.1.3 and updated the main text to provide additional information about the archaeological context of Chokhopani. The section formerly at the bottom of page 14 now reads:

page 15-16- “Instead, we hypothesize that malaria cases at highland sites may reflect transregional transmission from lowland areas capable of sustaining endemic foci. Situated in a high transverse Himalayan valley linking the Tibetan Plateau with southern lowland areas, the region surrounding Chokhopani may have served as an epicenter of trade and exchange in the first millennium BCE. Consisting of a series of shaft tombs built into a riverside cliff, the site contained three burial chambers harboring the remains of at least 21 individuals, as well as copper grave goods similar to those produced in the Indian subcontinent (Simons et al. 1994;

Aldenderfer 2013; Tiwari 1984; **Supplementary Note 1.1.3**)”

Comment 4: *Additionally, on p. 15, line 376, sex and age estimations at a minimum should be included in the text for the individual with *P. vivax*. One should not have to dig through the supplemental information to learn the person is genetically male and still without any indication of age. Without even basic information about this individual, the statement at the end of the paragraph that their disease status adds to this person’s osteobiography is overblown.*

We agree with the reviewer that more information about the malaria-infected individual from Chokhopani should be included in the main text. Unfortunately, due to the commingled nature of the remains, the only skeletal element available for analysis that can be securely attributed to the individual CHO001 is the tooth from which we obtained *P. falciparum* DNA (note that the statement in the main text that this individual was infected with *P. vivax* was a typo and has now been corrected). As such, we have very little data for this individual beyond their identification as a genetically male adult. With such limited information, we agree with the reviewer that it is inappropriate to make statements about this individual’s osteobiography. Instead, we have updated the text to demonstrate how our dataset adds to a picture of this person’s life that is otherwise very limited given the challenging archaeological context:

*page 16-17: “Due to the commingled nature of the remains, skeletal material from CHO001 is limited to the permanent molar yielding *P. falciparum* DNA. Previous studies found that the genetically male individual CHO001 possessed alleles associated with high-altitude adaptation and exhibited ancestry similar to present-day Tibetans (**Supplementary Note 1.1.3**; Liu *et al.* 2022). Notably, ancient individuals from Chokhopani also harbor a minor lowland South Asian ancestry component that is absent in other prehistoric sites in Upper Mustang; this finding further supports the connection between Chokhopani and lowland South Asian regions, although the admixture event likely occurred c. 500-1000 years before the *P. falciparum*-infected individual identified here lived (Liu *et al.* 2022) ... Overall, we highlight CHO001 as a rare case study in which aspects of an individual’s mobility can be inferred from their infectious disease status, an important finding given the limited information that could be drawn from the fragmented skeletal material associated with this individual.”*

Comment 5: *The possibility that this person [individual CHO001] was an immigrant from a nearby endemic area rather than a resident who traveled to such an area and returned should also be considered on p. 15, lines 381-382.*

We agree with the alternative hypothesis presented by the reviewer and have updated our statement on page 16 to reflect this additional possibility:

page 16- “Given the genetic links between CHO001 and other modern and ancient high-altitude populations, we suggest that this individual lived locally and contracted malaria while traveling to an adjacent endemic region; however, we cannot exclude the possibility that CHO001 was a recent immigrant who traveled to Chokhopani from a nearby endemic area.”

Comment 6: *At St. Rombout, in Mechelen, Belgium, on p. 16, line 412 on, what does co-infection with more than one species of Plasmodium suggest in terms of impact on the infected individuals, mobility, etc.? A little more explanation is needed.*

We agree with the reviewer that more information regarding malaria co-infection is needed to better interpret the results from Mechelen. Firstly, we note that all of the individuals with multi-species coinfections display genetic affinity to Mediterranean groups, suggesting the possibility of a non-local ancestry. We also add a brief statement to indicate that malaria coinfections are common today in

regions impacted by multiple *Plasmodium* spp. This is particularly relevant in light of the fact that *P. falciparum* is not thought to have been endemic north of the Alps, further supporting the idea that these individuals are non-local. These observations are made more explicit in the following updated text:

page 18-19- “Interestingly, we identified *P. vivax*, *P. malariae*, and/or *P. falciparum* in eight mid/late phase males, including three cases of multispecies *Plasmodium* infections, which are common today in geographic regions harboring more than one endemic species (**Supplementary Table 1**; Mayxay *et al.* 2004) ... Remarkably, all individuals infected with *P. falciparum* and/or *P. malariae*, including the three individuals with multispecies *Plasmodium* infections, exhibit non-local ancestry. As low winter temperatures are thought to have restricted endemic *P. falciparum* foci north of the Alps (Newfield *et al.* 2017), these findings are consistent with the hypothesis that the mid/late-phase malaria-infected individuals from Mechelen may have been troops from the circum-Mediterranean region.”

Regarding the clinical implications of multispecies coinfection, while we agree with the reviewer that the individual impact is highly relevant in this context, a review of the literature suggests that this is a topic of active research in modern malariology with significant knowledge gaps remaining (for a review of this topic, see Mayxay *et al.* 2004). Thus, we prefer not to speculate too much about the effect that these multispecies infections may have had on the health of the ancient individuals presented here, beyond the observation that they were infected with more than one *Plasmodium* spp. at their time of death.

Comment 7: *At the end of the paragraph on p. 17 lines 424-426, the last statement suggests that malaria was transmitted by infected soldiers to the local population. Were there any mosquito species present to transmit P. falciparum in the Low Countries during this time? This critical vector information is provided for Chokhopani and Chachapoyas areas but is lacking for Mechelen. Do you mean to say here that the infected soldiers contracted it elsewhere and died without transmitting it via Anopheles mosquitos to the local populace? Please clarify.*

We thank the reviewer for highlighting this area for clarification. *Anopheles* spp. capable of transmitting malaria are still present across Europe today and were likely also present in the Middle Ages/Early Modern period. While it is certainly possible that the infected soldiers presented here represent isolated cases, the presence of a competent local vector raises the possibility that they were part of more regional outbreaks. More data is needed to differentiate between these two hypotheses, making it an exciting question for further study. We have updated the text to provide information about the local mosquito vectors and clarify these two alternative transmission scenarios:

page 19-20- “Notably, multiple anopheline vectors capable of transmitting *P. falciparum* and other malaria parasites persist in the Low Countries and other regions of Europe today (Newfield 2017); thus, while *P. falciparum*-infected individuals at Mechelen may represent isolated, recently-imported cases, it is also possible that they fell victim to more extensive local malaria outbreaks triggered by intense human mobilization in the socio-economic context of warfare.”

Comment 8: *This issue arises again in the Conclusions and Implications section, lines 434-436. Invoking “human mobility in spreading malaria” suggests it was transmitted to local people, rather than simply being carried by the infected person to the area and dying out due to absence of mosquito vectors for P. falciparum in the Himalayan highlands and temperate Europe. The latter conclusion is supported by the evidence provided but the authors do not demonstrate that malaria spread to others. I suggest rephrasing here to “..human mobility in carrying malaria within them to peripheries of endemic zones.” If I have misunderstood, please clarify.*

While the presence of European vector spp. capable of transmitting *P. falciparum* leaves open the possibility of a local outbreak at Mechelen, we agree with the reviewers' assessment that the current dataset does not provide evidence for ongoing transmission either at Mechelen or Chokhopani. Therefore, we have adopted the Reviewer's suggested phrasing in the Conclusions and Implications section.

Comment 9: *As a minor editorial note, recall that data are plural (datum is singular). There is disagreement on line 109 ("data underscore" not "underscores"), line 277 and 431 (both should be "data provide"), and line 698 (should read "data were"). I noticed a couple of instances of such disagreement in the supplemental information file as well but did not record them.*

We have corrected these cases throughout the text.

Comment 10: *On line 113, "case-study" should be two separate words without a hyphen.*

We corrected this error in the text.

Reviewer Reports on the First Revision:

Referees' comments:

Referee #1 (Remarks to the Author):

I am satisfied with the authors' responses to my previous comments.

Referee #2 (Remarks to the Author):

My comments have been addressed. I just have some minor recommendations in regards to some of the figures.

In Figure 2A I could not figure out where any Bhutan Samples might be. Perhaps my "where is Waldo" skills are lacking but I suggest improving this.

Figure 3: The West Africa and Central Africa colors are barely distinguishable Consider making the West African samples slightly more light and slightly more red (but still brownish)?

Extended data Figure 1a: In the legend the symbols and the place names seem slightly out of alignment. Also next to it, I'm again unsure where Bhutan is.

Extended data Figure 3a: Again, West Africa and Central Africa colors are barely distinguishable.

Extended data Figure 7: Here West Africa and Central Africa colors are distinguishable due to being split up on a map. For consistency though, If altering them elsewhere also alter them here.

Referee #3 (Remarks to the Author):

The manuscript has now substantially improved, and the authors have addressed all my comments with a detailed human ancestry analysis.

My only remaining comments are:

-STR016 belongs to R1b1a1b1a1a2 as the authors mention in their response, but according to Supplementary Note 10 he further belongs to the DF27 subclade, which displays highest frequencies in the Iberian Peninsula since the Bronze Age. This supports their assessment of a likely Iberian origin for this individual.

-In Supplementary Note 10, "Two individuals (STR138 and malaria-infected STR091) are found dispersed on a PC1-PC2 range where individuals from Medieval-Early modern C. Italy and Aegean Byzantine Anatolia are plotted". According to Extended Figure 6b, these two individuals should be STR199 and STR091.

Referee #4 (Remarks to the Author):

The revised manuscript has been strengthened by additional analyses, inclusion of further information on context and individuals sampled, and clarification of vague or misleading statements. I am pleased that the authors have resolved my concerns. I have no further suggestions for improvements.

Author Rebuttals to First Revision:

Point-by-Point Response to Reviewer Comments for MANUSCRIPT NUMBER 2023-07-12551B “Ancient *Plasmodium* genomes shed light on the history of human malaria”

We received 7 specific comments from four reviewers. We present our responses to each reviewer comment below.

Referee #1 (Remarks to the Author): *I am satisfied with the authors' responses to my previous comments.*

Referee #2 (Remarks to the Author): *My comments have been addressed. I just have some minor recommendations in regards to some of the figures.*

Comment 1: *In Figure 2A I could not figure out where any Bhutan Samples might be. Perhaps my “where is Waldo” skills are lacking but I suggest improving this.*

We have fixed this inconsistency by changing the fill color of the two Bhutan strains to gray.

Comment 2: *In Figure 3, the West Africa and Central Africa colors are barely distinguishable. Consider making the West African samples slightly more light and slightly more red (but still brownish)?*

We have changed the color of the West African strains in Figure 3 to a slightly redder color.

Comment 3: *In extended data Figure 1a, in the legend the symbols and the place names seem slightly out of alignment. Also next to it, I'm again unsure where Bhutan is.*

We have realigned the symbol and place names in Figure 1a. We also changed the fill color of the two Bhutan strains to gray, as in Figure 2a.

Comment 4: *In extended data Figure 3a, again, West Africa and Central Africa colors are barely distinguishable.*

We have modified the color of the West African strains to be consistent with main text Figure 3.

Comment 5: *In extended data Figure 7, West Africa and Central Africa colors are distinguishable due to being split up on a map. For consistency though, if altering them elsewhere also alter them here.*

We have altered the color of the West African strains in Figure 7 for consistency with main text Figure 3 and Extended Data Figure 3a.

Referee #3 (Remarks to the Author): *The manuscript has now substantially improved, and the authors have addressed all my comments with a detailed human ancestry analysis.*

Comment 1: *STR016 belongs to R1b1a1b1a1a2 as the authors mention in their response, but according to Supplementary Note 10 he further belongs to the DF27 subclade, which displays highest frequencies in the Iberian Peninsula since the Bronze Age. This supports their assessment of a likely Iberian origin for this individual.*

We thank the reviewer for raising this point. We have added the following sentence in Supplementary Note 10 to highlight this finding: “Notably, individual STR016 belongs to the DF27 subclade of the R1b1a1b1a1a2 haplogroup, which is common among present-day Iberian populations and has been present in the region since the Bronze Age.”

Comment 2: *In Supplementary Note 10, "Two individuals (STR138 and malaria-infected STR091) are found dispersed on a PC1-PC2 range where individuals from Medieval-Early modern C. Italy and Aegean Byzantine Anatolia are plotted". According to Extended Figure 6b, these two individuals should be STR199 and STR091.*

We have corrected the text as follows: “Two individuals (STR199 and malaria-infected STR091) are found dispersed on a PC1-PC2 range where individuals from Medieval-Early modern C. Italy and Aegean Byzantine Anatolia are plotted.”

Referee #4 (Remarks to the Author): *The revised manuscript has been strengthened by additional analyses, inclusion of further information on context and individuals sampled, and clarification of vague or misleading statements. I am pleased that the authors have resolved my concerns. I have no further suggestions for improvements.*